# Follicular CD8+ T cells in *Trypanosoma cruzi* infection: helpers or killers depending on the target B cell population

Yamila Gazzoni[1,2], Laura Almada[1,2], Julio C. Gareca[1,2], Carolina L. Montes[1,2], Eva V. Acosta-Rodríguez[1,2], Adriana Gruppi [1,2]*

**1** Departamento de Bioquímica Clínica, Facultad de Ciencias Químicas (FCQ), Universidad Nacional de Córdoba (UNC), Córdoba, Argentina, **2** Centro de Investigaciones en Bioquímica Clínica e Inmunología (CIBICI - CONICET), Córdoba, Argentina

* agruppi@unc.edu.ar

## Abstract

Follicular cytotoxic T (Tfc) cells are a distinct subset of CD8+ T cells predominantly localized in B cell follicles and their surrounding areas. These cells play important roles in supporting B cell responses and controlling pathogens through the elimination of infected cells. Although their involvement in immune-mediated diseases and tumors is well-documented, their role in parasitic infections remains largely unexplored. Through phenotypic and transcriptomic analysis, we identified a specialized Tfc population that transiently emerges during the acute phase of *Trypanosoma cruzi* infection. Tfc cells in this context were composed mainly of effector cells, peaked concurrently with plasmablasts, and preceded the germinal center response. They exhibited high expression of proteins associated with B cell help, inflammatory chemokine receptors, and transcription factors linked to effector functions. *In vitro* assays revealed that Tfc cells display dual functionality: they promote antibody secretion by naïve and stimuli-activated B cells, and they also exert cytotoxic activity against plasmablasts, the antibody-producing cells present during the acute phase, through Fas/FasL interactions. Altogether, these findings suggest that Tfc cells may contribute to the regulation of early antibody responses during *T. cruzi* infection by combining helper and cytotoxic functions.

## Author summary

*Trypanosoma cruzi*, the causative agent of Chagas disease, induces a chronic infection in non-lymphoid tissues, evading complete clearance despite robust immune responses. Both B cells and CD8+ T cells are pivotal in the host defense: B cells produce parasite-specific antibodies that neutralize and lyse the

**Data availability statement:** The datasets generated for this study can be found in the NIH repository under accession number PRJNA1234210 (https://www.ncbi.nlm.nih.gov/sra/PRJNA1234210).

**Funding:** Research reported in this publication was supported by the Agencia Nacional de Promoción Científica y Tecnológica (https://www.argentina.gob.ar/ciencia/agencia, grant PICT 2018-01494 to AG), the Secretaría de Ciencia y Tecnología de la Universidad Nacional de Córdoba (https://www.unc.edu.ar/ciencia-y-tecnología/, grant 33620190100020CB to AG), the Consejo Nacional de Investigaciones Científicas y Técnicas (https://convocatorias.conicet.gov.ar/financiamiento-de-proyectos/, grant PIP 2015112201501005060CO to AG), and the National Institute of Allergy and Infectious Diseases (https://grants.nih.gov/grants/funding/r01.htm, grant R01AI116432 to AG). The funders had no role in study design, data collection and analysis, decision to publish, or preparation of the manuscript.

**Competing interests:** The authors have declared that no competing interests exist.

pathogen, while CD8[+] T cells target infected cells to control the intracellular parasite. Our study explores the interactions between these cell populations and identifies a specialized subset of CD8[+] T cells, termed follicular cytotoxic CD8[+] T cells (Tfc), which has been described in recent years for its role in shaping the B cell response. We found that Tfc cells transiently emerge during the acute phase of *T. cruzi* infection, coinciding with the appearance of plasmablasts, the key antibody-producing cells. Our results suggest that Tfc cells influence the B cell response in multiple ways: they promote early antibody production while regulating the survival of antibody-producing plasmablasts.

## Introduction

*Trypanosoma cruzi,* the causative agent of Chagas disease, establishes a chronic infection characterized by persistent parasitism in non-lymphoid tissues. Despite robust adaptive immune responses, the parasite is rarely eradicated. B and CD8[+] T cells play a pivotal role in host resistance in animal models and human Chagas disease (reviewed in [1]). B cells produce parasite-specific neutralizing and lytic antibodies (Abs) [2], while CD8[+] T cells control intracellular parasite replication by recognizing infected cells. The significance of these immune responses is highlighted by studies demonstrating that the absence of B cells [3,4] or depletion of CD8[+] T cells leads to a higher parasite burden during the acute phase of infection and worsens disease progression in chronically infected hosts [1,5].

Using different mouse and parasite strains, we observed that the initial B cell response against *T. cruzi* is directed toward generating Ab-secreting cells, identified as plasmablasts, whose peak of response precedes the peak of the germinal center (GC) reaction [6,7]. Consequently, a substantial B cell response leads to the production of both parasite-specific and non-specific Abs, with parasite-specific IgG isotypes detectable in serum only after 18 days post-infection (dpi) [6]. Beyond Ab secretion, we reported that extrafollicular plasmablasts from *T. cruzi*-infected mice produce IL-17 [8] and exhibit higher PD-L1 expression compared to other mononuclear cells, modulating the T cell response during the infection [7].

We and others have reported that B cells influence CD8[+] T cell responses [3,9]. Specifically, we found that depleting B cells with anti-CD20 Ab reduces both the number and function of cytotoxic CD8[+] T cells, an effect that can be restored by IL-17A supplementation [10]. However, whether CD8[+] T cells influence the B cell response in *T. cruzi* infection remains unknown.

In recent years, a population of CD8[+] T cells, known as follicular cytotoxic CD8[+] T (Tfc) cells has been reported. These cells are characterized by their ability to collaborate with B cells while maintaining cytolytic properties [11,12]. Tfc cells are defined by the expression of CXCR5 and exhibit a profile of costimulatory molecules, transcription factors, inhibitory genes, and proteins similar to those of follicular helper CD4[+] T (Tfh) cells (reviewed by Valentine and Hoyer, [13]). In contrast to CXCR5[-]CD8[+] T cells, Tfc cells possess the ability to infiltrate B cell follicles under

conditions of chronic antigen exposure and inflammation. CXCR5+CD8+ T cells are found under diverse pathogenic conditions, promoting cell lysis in viral infections and cancers [14–18], and functioning as helper cells in inflammation and autoimmunity [19,20]. Importantly, their function seems to be shaped by the immune context, potentially varying across pathological conditions. However, their presence and role in protozoan infections, particularly in *T. cruzi* infection, remain unexplored.

In this study, we performed a detailed phenotypic and transcriptomic characterization of Tfc cells that emerge during the acute phase of *T. cruzi* infection and persist transiently. Through *in vitro* assays, we examined the interactions of Tfc cells with naïve and stimuli-activated B cells, as well as with differentiated Ab-secreting cells such as plasmablasts. Here, we describe that Tfc cells from *T. cruzi* infected mice exhibit a dual function: while they promote Ab secretion by naïve B cells via soluble factor/s they also can exert cytotoxicity on differentiated Ab-secreting cells through Fas-L.

## Results

### Tfc cell response in *T. cruzi* infection peaks concurrently with plasmablasts and precedes the germinal center response

As a first step in evaluating Tfc cells during *T. cruzi* infection, we assessed their presence in the spleens of infected mice at different dpi. CXCR5 and PD-1 coexpression within the CD8+ T cell population enabled the identification of Tfc cells, which were detected in the spleens of infected mice but were absent in non-infected controls (Fig 1A).

Kinetic analysis revealed that the frequency of splenic Tfc cells followed a similar pattern to that of plasmablasts (IgD- CD138+B220int) (Fig 1B). Both populations peaked around 18 dpi, preceding the GC B cell (Fas+GL-7+B220+) response. Accordingly, the frequency of Tfc cells declined as GC B cells began to increase. The gating strategy for B cell subsets is detailed in S1A Fig. In addition, we found that the kinetics of Tfc cells mirrored the parasitemia curve, which peaked around 18 dpi (Fig 1C).

Given that Tfc cells are defined as a subset of CD8+ T cells enriched in B cell follicles and their surroundings [21], we evaluated the presence of CD8+ cells within B cell follicles in infected mice. To this end, we performed immunofluorescence analysis of spleen sections from infected mice at 15, 18, and 23 dpi. As shown in Fig 1D, CD8+ cells (red) were largely excluded from B cell follicles (blue) in non-infected mice. By 15 dpi, when GCs began to form, as evidenced by PNA staining (green), CD8+ cells started to appear within B cell areas. CD8+ cells were observed within the GC area. At 18 dpi, spleen sections revealed CD8+ cells distributed throughout the B cell follicle, including within the GC itself, with higher-magnification views highlighting their localization. By 23 dpi, when GCs were well-formed and occupied most of the follicle, CD8+ cells were no longer detected in the B cell area. Although anti-CD8 staining did not distinguish Tfc cells from other CD8+ cells, their localization within the B cell follicular structure suggests that these cells could be Tfc cells, consistent with the flow cytometry results.

In contrast to the follicular localization described above, immunofluorescence analysis at 18 and 23 dpi (Fig 1E) revealed that, outside B cell follicles, Ab-secreting cells identified by anti-CD138 staining (green) were found in proximity to CD8+ cells (red). These observations suggest potential spatial interactions between these two populations.

### Tfc and Non-Tfc cells from *T. cruzi* infected mice exhibit distinct transcriptomic signatures

To better characterize Tfc cells, we determined their transcriptional profile by performing RNA sequencing on CXCR5+PD-1+CD8+ T cells isolated by cell-sorting from the spleens of 18 dpi-infected mice, along with CXCR5-PD-1-CD8+ T (Non-Tfc) cells obtained from the same samples. S1B Fig illustrates the gating strategy used for the sorting of Tfc and Non-Tfc CD8+ T cells. Principal Component Analysis revealed that Tfc and Non-Tfc cells displayed distinct transcriptomic profiles, segregating along PC1, which accounted for 78.57% of the variance (Fig 2A).

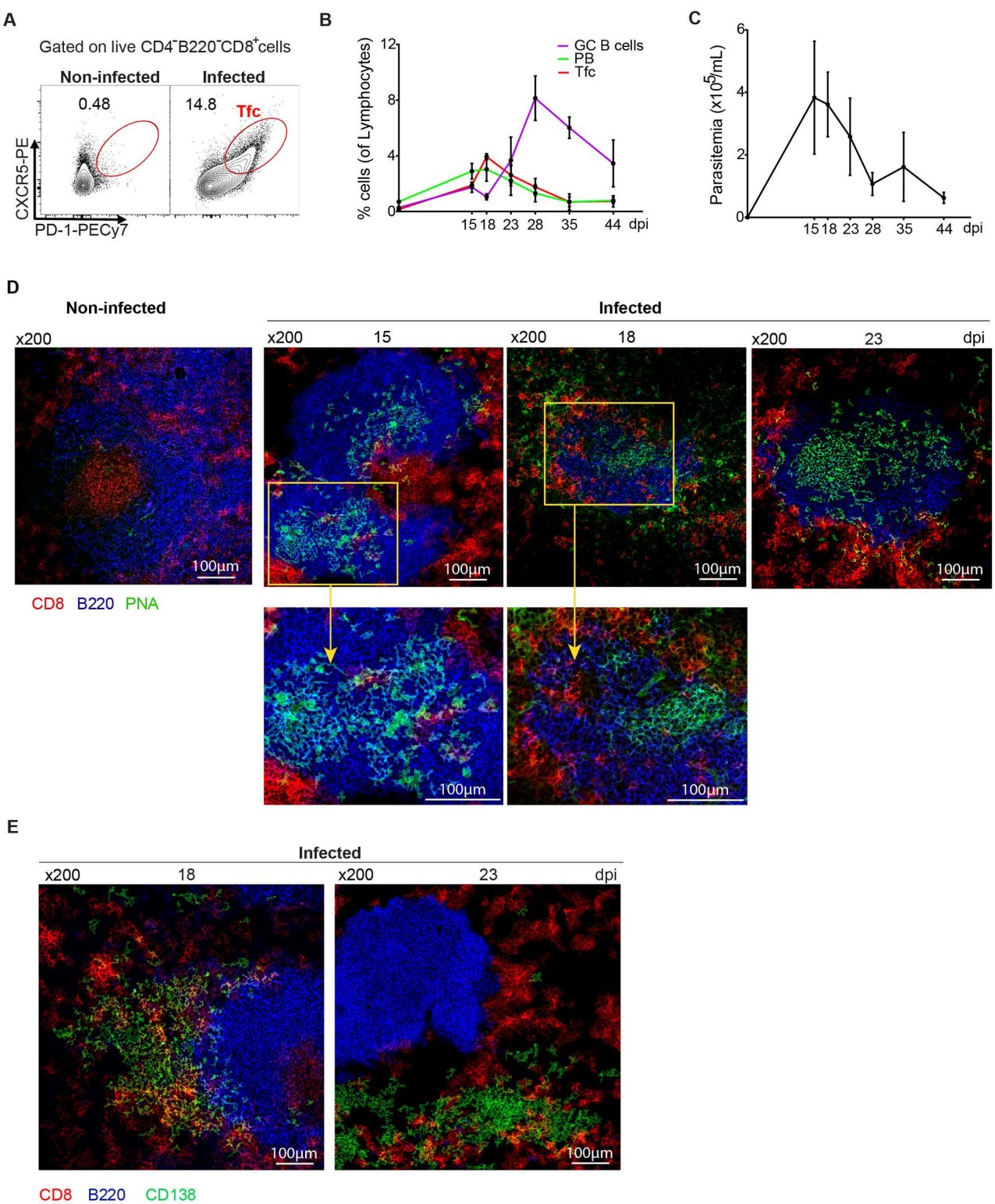

**Fig 1. Dynamics of Tfc cells, GC B cells and plasmablasts in the spleens of *T. cruzi*-infected mice.** C57BL/6 mice were intraperitoneally injected with PBS (non-infected) or with 5000 trypomastigotes of *T. cruzi* Tulahuén strain (infected). Splenic lymphocyte populations were analyzed by flow cytometry at different days post-infection (dpi). **(A)** Representative contour plot showing the percentage of Tfc cells (CXCR5+PD-1+) gated on live

CD4-B220-CD8+ T lymphocytes in the spleen of 18 days-infected mice and control (non-infected). **(B)** Kinetics of the frequency of GC B cells (purple; Fas+GL-7+B220+), plasmablasts (PB, green; IgD-CD138-B220int), and Tfc cells (red), expressed as a percentage of total lymphocytes. **(C)** Parasite counts in blood (parasitemia, black) during infection. **(B, C)** Data are presented as mean±SD; N=3-4 mice per dpi. **(D)** Immunofluorescence images of spleen sections (7 μm) from non-infected and *T. cruzi*-infected mice at 15, 18, and 23 dpi, stained with PNA (green), anti-B220 (blue), and anti-CD8 (red). Yellow insets show ×2 magnified views of the boxed areas. **(E)** Immunofluorescence of spleen sections harvested at 18 and 23 dpi, stained with anti-CD138 (green), anti-B220 (blue), and anti-CD8 (red). Magnification: ×200. Scale bar: 100 μm. N=3-4 mice per dpi. Data are representative of two (A-C) and three (D-E) independent experiments.

Analysis of differentially expressed genes identified 1,252 transcripts differentially expressed between the two cell populations. Among these, 732 genes were significantly upregulated in Tfc cells (orange dots), while 520 were downregulated (green dots) compared to Non-Tfc cells (Fig 2B).

We analyzed the expression patterns of gene sets related to B cell collaboration, chemokine receptors, transcription factors, cytokines, and CD8+T cell effector molecules, summarizing the results in a heatmap (Fig 2C). As expected, and consistent with their ability to interact with B cells [22], Tfc cells were enriched in transcripts encoding molecules typically expressed by Tfh cells, a specialized subset of CD4+T cells that provide essential help to B cells [23], including *Icos, Bcl6, Pdcd1, Tnfrsf9, Tnfrsf4,* and *Batf*. They also showed a distinct expression profile of *Ccr7* and several chemokine-encoding transcripts: compared to Non-Tfc cells, Tfc cells exhibited higher levels of *Ccl3, Ccl4, Ccl8, Cxcl9, Cxcl10, Cxcl16,* and *Xcl1*, while expressing lower levels of *Ccl5* and *Ccl9*.

In addition, Tfc cells from *T. cruzi*-infected mice exhibited increased expression of genes critical for CD8+T cell activation, proliferation, and differentiation, such as *Mtor* [24], involved in metabolic programming; *Mki67*, a marker of proliferation [25]; and *Irf4*, essential for T cell differentiation [26]. In contrast, *Tcf7*, associated with memory T cell formation, was more highly expressed in Non-Tfc cells. Notably, Tfc cells also showed increased expression of *Il21*, a cytokine essential for Tfh cell function [27], as well as *Ifng* and *Tnf*, while Non-Tfc cells expressed higher levels of *Gzmk, Gzma,* and *Gzmm*.

Gene Set Enrichment Analysis (GSEA) revealed a significant enrichment of genes associated with *Immunoglobulin Production* and *Positive Regulation of B Cell Activation* in Tfc cells compared to Non-Tfc cells (Fig 2D). Additionally, Gene Ontology (GO) enrichment analysis showed a significant overrepresentation of biological processes associated with immune functions. Among the most enriched GO terms were *innate immune response*, *phagocytosis* (engulfment and recognition), *positive regulation of B cell activation*, *complement activation, antigen processing and presentation via MHC class II,* and *response to interferon-gamma* (S2 Fig).

Together, these findings support the identification of CXCR5+PD-1+CD8+ T cells as Tfc cells and demonstrate that Tfc and Non-Tfc cells in the spleen of *T. cruzi*-infected mice constitute transcriptionally distinct populations.

## Tfc cells express high levels of proteins related to B cell help, inflammatory chemokine receptors and transcription factors linked to effector functions

To further characterize Tfc and Non-Tfc cells in the context of *T. cruzi* infection, we used multiparametric flow cytometry to assess the expression of various surface and intracellular proteins. The gating strategy shown in Fig 3A was used to define Tfc (CXCR5+PD-1+CD8+) and Non-Tfc (CXCR5-PD-1-CD8+) populations. These gates were used for all subsequent phenotypic and functional analyses. We first focused on molecules involved in B cell help, such as ICOS, CD40L, and Bcl6. Tfc cells from acutely infected mice exhibited significantly higher expression levels of all these markers, measured as mean fluorescence intensity (MFI), compared to their Non-Tfc counterparts, as shown in representative histograms and cumulative data (Fig 3A).

Given the key role of ICOS and CD40L in T–B cell cooperation [28], we compared their expression in Tfc cells and the canonical B cell helper population, Tfh cells (gating strategies detailed in S1C Fig). As expected, Tfc cells expressed lower levels of ICOS than Tfh cells, while CD40L expression was similar between the two populations (Fig 3B), highlighting the capacity of Tfc cells to support B cell responses.

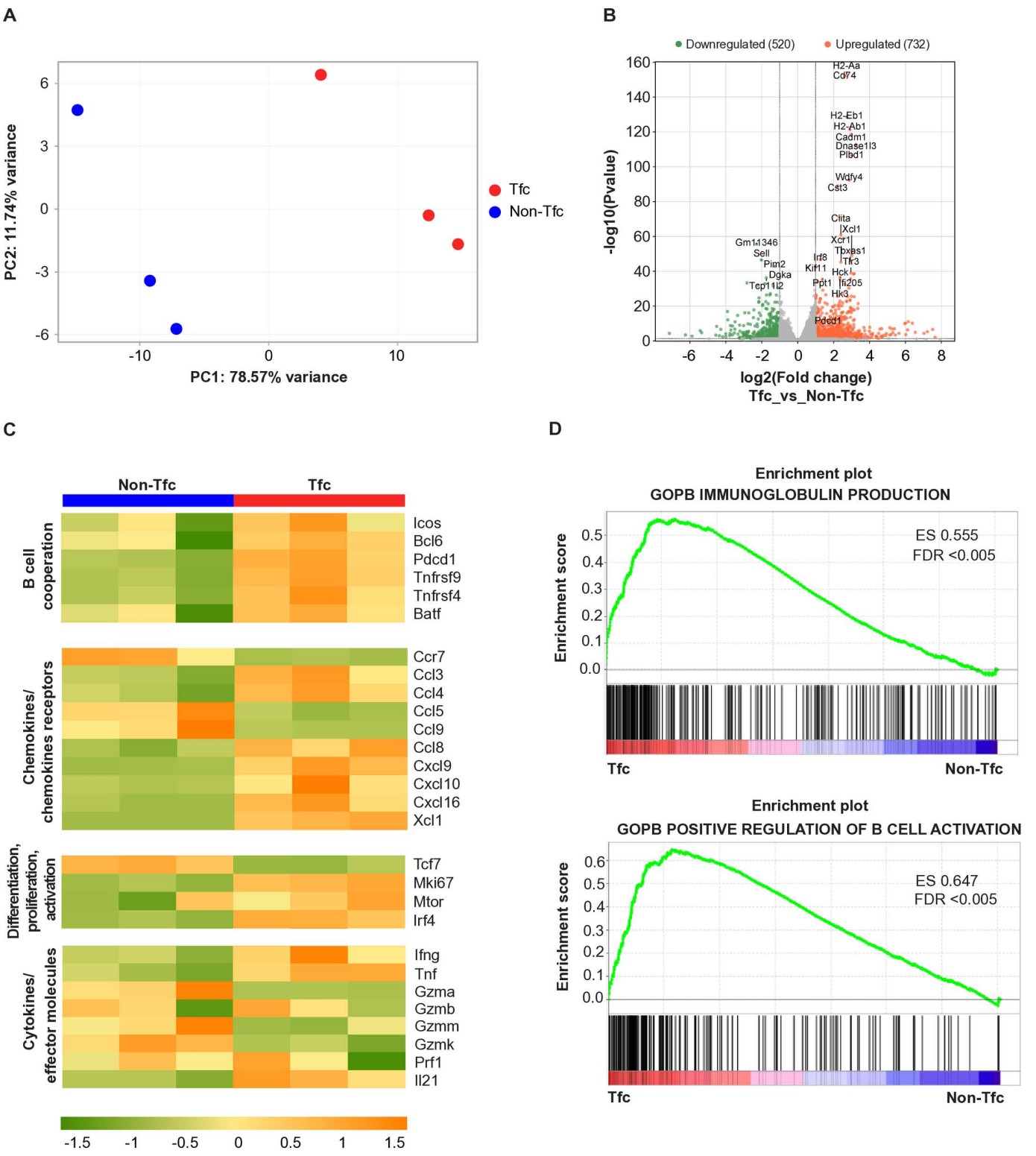

**Fig 2. Transcriptional analysis of Tfc and Non-Tfc cells from *T. cruzi* infected mice.** CXCR5⁺PD-1⁺CD8⁺T cells (Tfc) and CXCR5⁻PD-1⁻CD8⁺ (Non-Tfc) T cells were isolated by cell sorting from the spleens of *T. cruzi*-infected mice obtained at 18 dpi for RNA sequencing. **(A)** Principal Component (PC) Analysis plot of the whole transcriptome of Tfc (red) and Non-Tfc cells (blue). Each circle represents a biological replicate, where each replicate corresponds to a pool of three mice. **(B)** Volcano plot displaying differentially expressed genes between Tfc and Non-Tfc cells. Significantly upregulated

(orange dots) and downregulated (green dots) genes were defined by log2 fold change >1 and adjusted p-value < 0.05. Selected genes are labeled. **(C)** Heatmap showing the relative expression of selected genes in Tfc and Non-Tfc cells. The color scale represents the Z-score of gene expression, with green indicating lower expression and orange higher expression. **(D)** Gene Set Enrichment Analysis (GSEA) plots highlighting two enriched pathways related to B cell responses, identified using the M5 (Gene Ontology sets) mouse collection from MSigDB. ES: enrichment score; FDR: false discovery rate. In the plots, genes most enriched in Tfc cells are distributed toward the left, while genes with lower enrichment, predominant in Non-Tfc cells, are found toward the right.

We next analyzed the expression of chemokine receptors involved in cell positioning within B cell follicles, as well as three transcription factors: T-bet, Eomes, and TCF-1, which regulate CD8$^+$T cell differentiation and function [29,30]. Representative histograms and MFI data revealed that Tfc cells displayed higher expression levels of CXCR3, CXCR4, and CCR7 (Fig 3C), as well as of the three transcription factors, compared to Non-Tfc cells (Fig 3D).

### Tfc cells from *T. cruzi* infected mice are composed predominantly of effector cells

To identify potential roles of Tfc cells in *T. cruzi* infection, we first assessed various cellular markers associated with distinct differentiation states and functions. To this end, we used the gating strategy shown in Fig 3A to define Tfc (CXCR5$^+$PD-1$^+$) and Non-Tfc (CXCR5$^-$PD-1$^-$) CD8$^+$populations. Based on the expression of CD44 and CD62L, we analyzed the frequency of cells with effector/effector memory (CD44$^+$CD62L$^-$), central memory (CD44$^+$CD62L$^+$), or naïve (CD44$^-$CD62L$^+$) phenotypes within Tfc and Non-Tfc subsets. Tfc cells predominantly exhibited effector and effector memory phenotypes, with a smaller proportion displaying a central memory phenotype. As expected, Non-Tfc cells included a broader range of differentiation states, from naïve to effector and memory cells (Fig 4A).

Given that Non-Tfc population contains a subset of naïve T cells, we reanalyzed the protein expression data shown in Fig 3 by excluding naïve CD8$^+$T cells. We then compared Tfc cells to CD44$^+$ Non-Tfc cells (S3 Fig). Even after excluding naïve cells from the Non-follicular population, Tfc cells, all of which are CD44$^+$, exhibited higher levels of ICOS, CD40L, and Bcl6 (S3B Fig); CXCR3, CXCR4, and CCR7 (S3C Fig); as well as T-bet, Eomes, and TCF-1 (S3D Fig) compared with CD44$^+$Non-Tfc cells. Taken together, the higher expression of those proteins in Tfc from *T. cruzi*-infected mice is not related to the activation status of CD8$^+$T cells, since activated CD44$^+$Non-Tfc CD8$^+$T cells expressed significantly lower levels of these molecules compared to Tfc cells.

We showed that CCR7 expression was reduced in Non-Tfc cells from infected mice, despite this population including a subset of naïve CD8$^+$T cells that typically express high levels of CCR7 and TCF-1 [31,32]. To evaluate whether this pattern could be explained by infection, we compared CCR7 and TCF-1 expressions in Tfc and Non-Tfc cells from infected mice with that in naïve CD8$^+$T cells from non-infected controls. As shown in S3E Fig, naïve CD8$^+$T cells from non-infected mice (CD8$^+$ naïve NI) expressed CCR7 and TCF-1, and this expression was higher than in Non-Tfc cells from infected mice. Within the Tfc subset, the frequency of CCR7$^+$ and TCF-1$^+$cells was also reduced; however, CCR7 MFI was slightly higher than that observed in naïve CD8$^+$T cells from non-infected mice, while TCF-1 levels remained comparable.

Interestingly, compared to Non-Tfc cells, a higher frequency of Tfc cells recognized the immunodominant *T. cruzi* peptide TSKB20, suggesting that Tfc cells are enriched in parasite-specific cells (Fig 4B). Consistently, in *in vitro* cultures where splenic cells were stimulated with PMA/Ionomycin or TSKB20, we detected a higher frequency of CD107$^+$, IFN-γ$^+$, IL-6$^+$, IL-21$^+$, IL-10$^+$ and IL-4$^+$ cells within Tfc cells compared to Non-Tfc cells (Fig 4C, gating strategies detailed in S1D Fig). Additionally, a higher frequency of TNF-producing Tfc cells was also observed upon TSKB20 stimulation. However, no differences were observed in the frequencies of Granzyme B$^+$ (GrzB$^+$) or Perforin$^+$ (Prf$^+$) cells between Tfc and Non-Tfc cells under either stimulation condition.

To further investigate the effector molecules expressed by Tfc cells, we next evaluated the cytokine secretion profile of Tfc and Non-Tfc cells purified from the spleens of mice at 18 dpi. Sorted Tfc and Non-Tfc cells were stimulated with anti-CD3 plus anti-CD28 for 4 h, and secreted cytokines were quantified using a multiplex bead-based assay. We found

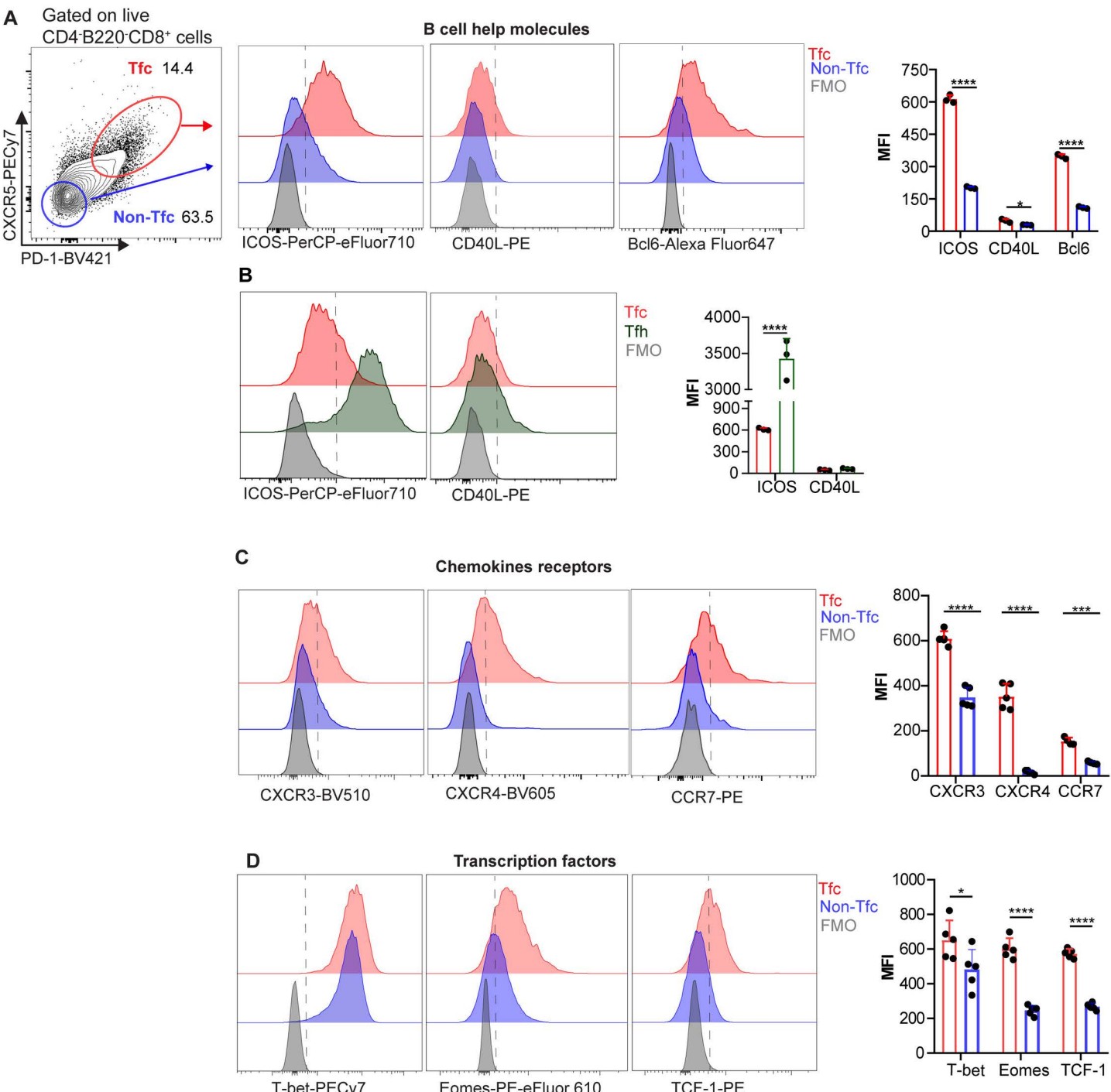

**Fig 3. Expression of B cell-help molecules, chemokine receptors and transcription factors in Tfc cells.** Splenic cells from *T. cruzi* infected mice were collected at 18 dpi and analyzed by flow cytometry. **(A)** Representative contour plot (left) showing Tfc and Non-Tfc gating strategy, used for the analyses in panels A, C, and **D. (A-D)** Representative histograms and statistical analysis of mean fluorescence intensity (MFI) of **(A-B)** B cell help molecules (ICOS, CD40L, and Bcl6), (C) chemokine receptors (CXCR3, CXCR4, and CCR7), and (D) transcription factors (T-bet, Eomes, and TCF-1) in Tfc and Non-Tfc cells (A, C, D) or in Tfc and Tfh cells **(B)**. CD40L expression was assessed on the cell surface. **(A-D)** Data are presented as mean±SD; N=3-5 mice. **(A-D)** Data are representative of 3 independent experiments. Statistical significance was determined by an unpaired *t*-test (*p<0.05, **p<0.01, ***p<0.001, ****p<0.0001).

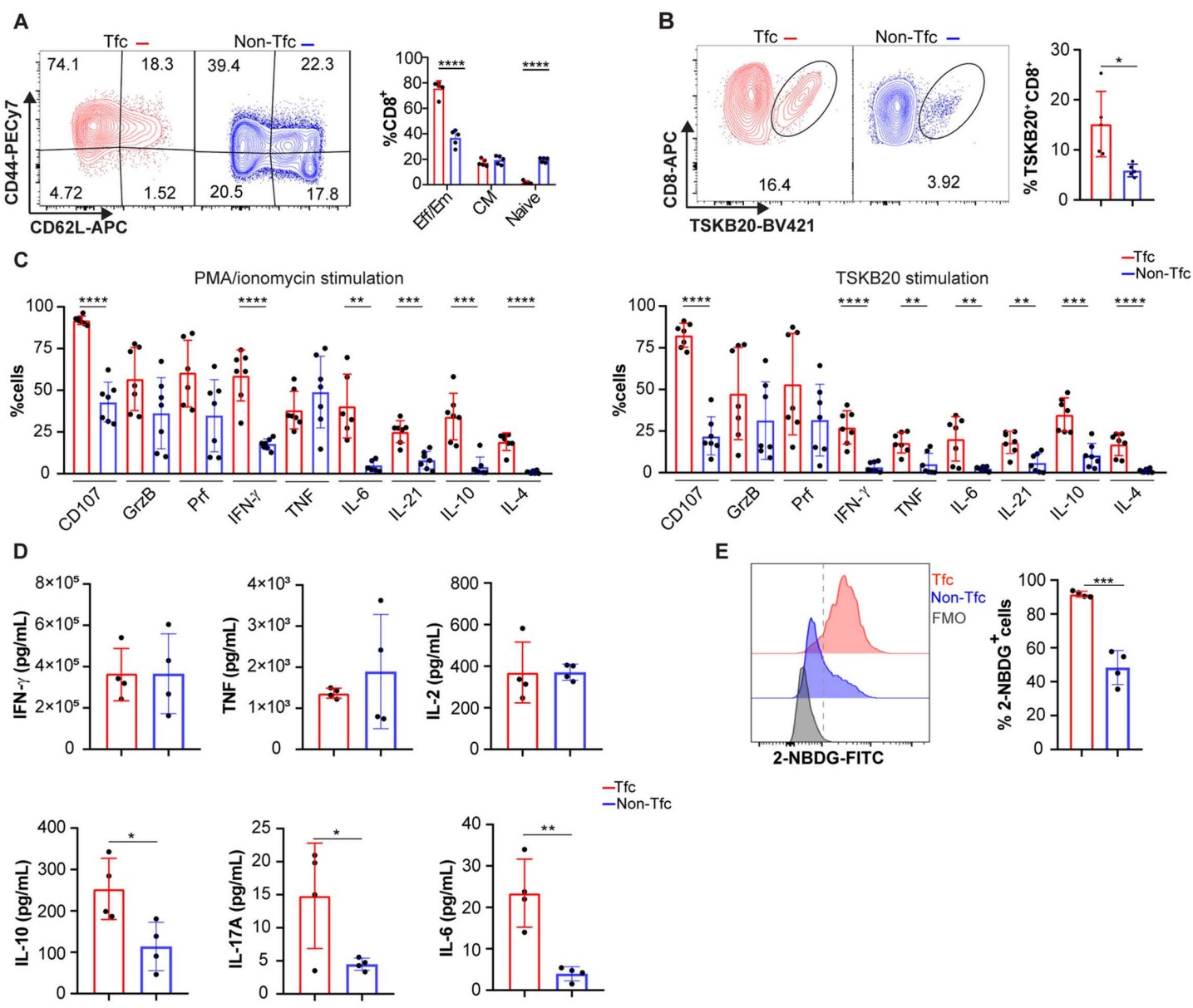

**Fig 4. Phenotypic and functional characterization of Tfc and Non-Tfc cells.** Splenic cells from *T. cruzi* infected mice were collected at 18 dpi and analyzed by flow cytometry. **(A-B)** Representative contour plots and statistical analysis of the frequency of (A) naïve (CD44⁻CD62L⁺), effector/effector memory (Eff/Em, CD44⁺CD62L⁻) and central memory (CM, CD44⁺CD62L⁺) cells; and **(B)** TSKB20⁺CD8⁺cells among gated Tfc and Non-Tfc cells. **(C)** Statistical analysis of the frequency of CD107⁺ Tfc or Non-Tfc cells and cytokine-producing cells (Granzyme B, Perforin, IFN-γ, TNF, IL-6, IL-21, IL-10, and IL-4) following 4 h stimulation with either PMA/ionomycin (left) or the TSKB20 peptide (right). **(D)** Statistical analysis of IFN-γ, TNF, IL-2, IL-6, IL-10, and IL-17A concentrations in culture supernatants of sorted Tfc and Non-Tfc cells following 4 h stimulation with anti-CD3 plus anti-CD28, assessed using a multiplex bead-based assay. **(E)** Representative histogram and statistical analysis of the percentage of 2-NBDG⁺ Tfc and Non-Tfc cells. **(A-E)** Data are presented as mean ± SD. (A, B, D, **E**) N = 4-5 mice and **(C)** N = 7. Data are representative of 3 **(A-B)**, or 2 (C-E) independent experiments. Statistical significance was determined by an unpaired *t*-test (*p < 0.05, **p < 0.01, ***p < 0.001, ****p < 0.0001). Abbreviations: GrzB (Granzyme **B**), Prf (Perforin).

that Tfc and Non-Tfc cells secreted comparable levels of IFN-γ, TNF, and IL-2. In contrast, Tfc cells produced significantly higher amounts of IL-10, IL-17A, and IL-6 compared to Non-Tfc cells (Fig 4D).

Finally, consistent with their effector activity profile [33,34], a greater proportion of Tfc cells exhibited glucose uptake compared to Non-Tfc cells (Fig 4E), as assessed using the fluorescent D-glucose analog 2-NBDG [35].

Altogether, these phenotypic and functional features of Tfc cells arising during acute *T. cruzi* infection indicate that this population exhibits effector characteristics associated with both B cell help and conventional CD8+ T cell functions.

## Tfc cells promote class-switched Ab secretion by naïve B cells through soluble factors

To investigate the role of Tfc cells in *T. cruzi*-infected mice, considering their phenotypic and effector characteristics and their reported collaboration with B cells, we initially performed co-culture assays with naïve B cells. These assays aimed to assess phenotypic changes and immunoglobulin (Ig) secretion. Fig 5A shows a scheme of the co-culture system, in which naïve splenic B cells (IgD+IgM+B220+) from *T. cruzi*-infected mice were incubated with either Tfc or Non-Tfc cells, all purified from the spleens of mice at 18 dpi by cell sorting. T cells were activated using anti-CD3 plus anti-CD28.

After 20 h of interaction, both CD8+ T cell populations induced an expansion of IgD- B220+ B cells, but the effect was significantly greater in cultures with Tfc cells. Flow cytometry analysis revealed that both Tfc and Non-Tfc cells enhanced the expression of the activation markers Fas and MHC-II, as measured by MFI. Additionally, they induced a slight but significant increase in the transcription factor Blimp1, which is crucial for B cell differentiation into Ab-secreting cells (Fig 5B).

A multiplex bead-based assay demonstrated that Tfc cells promoted the secretion of IgM and class-switched Igs, including IgA, IgG2b, IgG2c, and IgG3 (Fig 5B). Notably, IgG2c, the predominant isotype found in the serum of *T. cruzi*-infected mice [36] was detected at the highest concentration in the culture supernatants. Similar results were obtained when naïve B cells from non-infected mice were used (S4 Fig). These findings indicate that although both CD8+ T cell subsets can activate B cells, only Tfc cells effectively promote Ig secretion.

Interestingly, when naïve B cells and CD8+ T cell subsets, all derived from infected mice, were co-cultured with the parasite peptide TSKB20, which is recognized by T cells, similar findings were observed (Fig 5C). Specifically, an increased secretion of IgG2c, along with IgM, IgA, and IgG2b which were detected exclusively in the supernatants of B cells incubated with Tfc cells stimulated by TSKB20. Non-Tfc cells only slightly increased the production of IgG2c.

To compare the B cell helper capacity of Tfc cells with that of Tfh cells, the canonical B cell helper subset [37,38], we performed co-culture assays using sorted Tfh and naïve B cells from the spleens of *T. cruzi*-infected mice at 18 dpi. After 20 h of culture, Tfh CD4+ T cells efficiently induced Ab production, reaching levels comparable to those induced by Tfc cells. Notably, IgM and IgG2c concentrations were significantly higher in the supernatants of cultures containing Tfh cells (S5A Fig).

To determine whether Ig production driven by Tfc cells depends on soluble mediators or cell contact, we performed co-culture assays using transwell inserts. The transwell system significantly reduced the MFI of Fas and MHC-II on B cells cultured with anti-CD3 plus anti-CD28 activated CD8+ T cell subsets, indicating that cell-to-cell contact is required for B cell activation (Fig 5D). Even in the transwell setup, Tfc cells effectively promoted the secretion of all Ig isotypes at levels comparable to those observed without the transwell, except for IgM, which showed a slight reduction (Fig 5D).

To identify possible soluble mediators involved in the Tfc cell helper function, we investigated the role of IFN-γ by adding a neutralizing anti-IFN-γ Ab to the co-cultures. IFN-γ was selected because it is abundantly produced by Tfc cells, and a higher proportion of Tfc population produces IFN-γ compared to Non-Tfc cells (Fig 4C). The blockade resulted in a significant reduction in the concentration of multiple Ig isotypes, while IgG1 and IgG3 remained unaffected (Fig 5E).

## Tfc cells increase the production of certain Ig isotypes by stimuli-activated B cells

We next examined the impact of Tfc cells on activated B cells. As shown in Fig 6A, naïve B cells from non-infected mice were activated *in vitro* using a combination of CpG and anti-CD40, a commonly employed stimulus to promote B cell

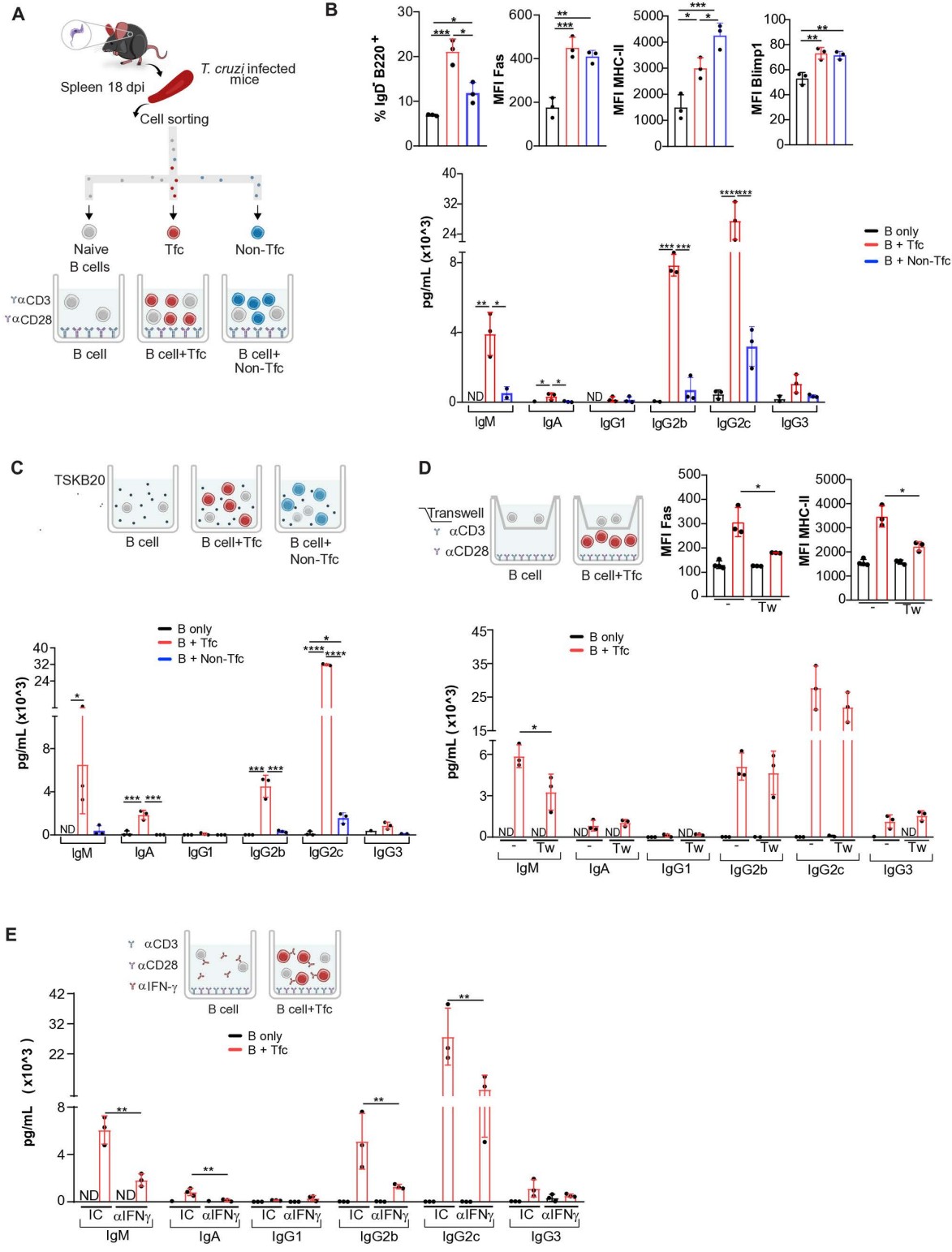

**Fig 5. Activation and Ab production by naïve B cells in the presence of Tfc and non-Tfc cells. (A-B)** Sorted naïve B cells were co-cultured for 20 h with medium alone (black), with sorted Tfc cells (red), or sorted Non-Tfc cells (blue), all purified from 18-dpi infected mice. **(A)** Schematic representation of the *in vitro* culture assay. **(B)** Statistical analysis of the frequency of IgD⁻B220⁺ cells and the MFI of Fas, MHC-II, and Blimp1 in B cells (upper panel) and statistical analysis of Ig concentrations (IgM, IgA, IgG1, IgG2b, IgG2c, and IgG3) measured in the culture supernatants using a multiplex

bead-based assay (lower panel). **(C)** Schematic design of the co-culture of B cells with Tfc and Non-Tfc stimulated with TSKB20; and statistical analysis of Igs concentrations in the co-culture supernatants. **(D)** Co-culture described in (A) were performed in the presence or absence of a transwell system. The upper panel shows a schematic representation of the *in vitro* culture model and statistical analysis of MFI of Fas and MHC-II in B cells. The lower panel presents the statistical analysis of Ig concentrations in co-culture supernatants. **(E)** Co-culture described in (A) were performed with anti-IFN-γ, and an isotype control (IC) and Ig concentrations were determined in supernatants. ND: not detected. For statistical analysis, samples with ND values were imputed with a value equal to the lower detection limit of the technique employed. Flow cytometry data were analyzed on CD8⁻B220⁺cells. Each dot represents an individual animal. Data are representative of 2 (D-E) and 4 (A-C) independent experiments. Statistical analyses: ordinary one-way ANOVA with selected comparisons and Bonferroni correction **(A-C)**; unpaired *t*-test **(D-E)**. *p < 0.05, **p < 0.01, ***p < 0.001, ****p < 0.0001.

activation and differentiation [39,40]. After 24 h, stimuli-activated B cells were washed and co-cultured with sorted Tfc and Non-Tfc cells from mice at 18 dpi, previously stimulated with either anti-CD3 plus anti-CD28 or TSKB20.

After 20 h of co-culture, both Tfc and Non-Tfc cells, regardless of the T cell stimuli, increased the frequency of Blimp1⁺B220⁺cells (Fig 6B), suggesting enhanced differentiation into Ab-secreting cells. As expected, supernatants from stimuli-activated B cells contained high concentrations of IgM and IgA [41–43], while IgG1, IgG2b, and IgG3 were barely detectable. When stimuli-activated B cells were incubated with Tfc cells stimulated with anti-CD3 plus anti-CD28, IgM and IgG2c levels increased significantly, along with a modest but significant increase in IgG3 (Fig 6C). In contrast, when TSKB20 was used to stimulate Tfc cells, only IgG2c levels increased in the supernatant of co-cultures with stimuli-activated B cells (Fig 6D).

These findings indicate that both CD8⁺T cell subsets from *T. cruzi*-infected mice can enhance B cell differentiation into Ab-secreting cells. However, only Tfc cells promoted the secretion of certain Ig isotypes by stimuli-activated B cells.

## Tfc cells reduce Igs secretion by plasmablasts via Fas/FasL-induced cell death

Given the proximity between CD8⁺ T cells and extrafollicular plasmablasts observed in immunofluorescence assays (Fig 1E), we hypothesized that CD8⁺ T cells might modulate Ab-secreting cells function. To test this, we co-cultured sorted splenic plasmablasts (IgD⁻CD138⁺B220ⁱⁿᵗ) with either Tfc or Non-Tfc cells (schematized in Fig 7A). The presence of either CD8⁺T cell subset led to a marked reduction in Ig secretion by plasmablasts, with significant reductions in IgM, IgA, IgG1, IgG2b, IgG2c, and IgG3 concentrations in the co-culture supernatants (Fig 7B). Notably, this reduction correlated with a significant decline in plasmablast viability in co-cultures with both Tfc and Non-Tfc cells (Fig 7B). In contrast, when a transwell system was used (Fig 7C), neither Ig concentrations nor plasmablast viability were affected, indicating that direct cell-to-cell contact was required to mediate these effects.

To investigate whether the Fas/FasL pathway was involved, we assessed FasL expression in Tfc and Non-Tfc cells and Fas expression in plasmablasts and naïve B cells. As shown in Fig 7D, Tfc cells expressed significantly higher levels of FasL than Non-Tfc cells, and plasmablasts expressed more Fas than naïve B cells. Based on these findings, we performed co-cultures in the presence of a blocking anti-FasL Ab. Despite the different expression of FasL between both subsets of CD8⁺ T cells, blockade of the Fas–FasL pathway restored Ig secretion to levels comparable to those observed in plasmablast-only cultures and fully prevented the loss of plasmablast viability triggered by Tfc and Non-Tfc cells (Fig 7E). These results indicate that CD8⁺T cells from *T. cruzi*-infected mice impair Ab production by inducing plasmablast death through Fas–FasL-mediated cytotoxicity.

Given that Tfc cells share phenotypic and functional features with Tfh cells, we investigated whether Tfh cells might also exert regulatory functions on differentiated B cells, such as plasmablasts. We first assessed FasL expression in Tfh cells and found that, while both Tfc and Tfh cells expressed FasL, the levels were lower in Tfh cells (S5B Fig). To directly compare their regulatory capacity, we performed co-culture assays using sorted Tfc, Tfh, and plasmablasts isolated from the spleens of *T. cruzi*-infected mice at 18 dpi. All cell populations were activated for 20 h with anti-CD3 plus anti-CD28. In these co-cultures, we observed reduced concentrations of multiple Ig isotypes in the supernatants of plasmablasts cultured with Tfc or Tfh cells, indicating a suppression of Ab secretion. Notably, this inhibitory effect was more pronounced

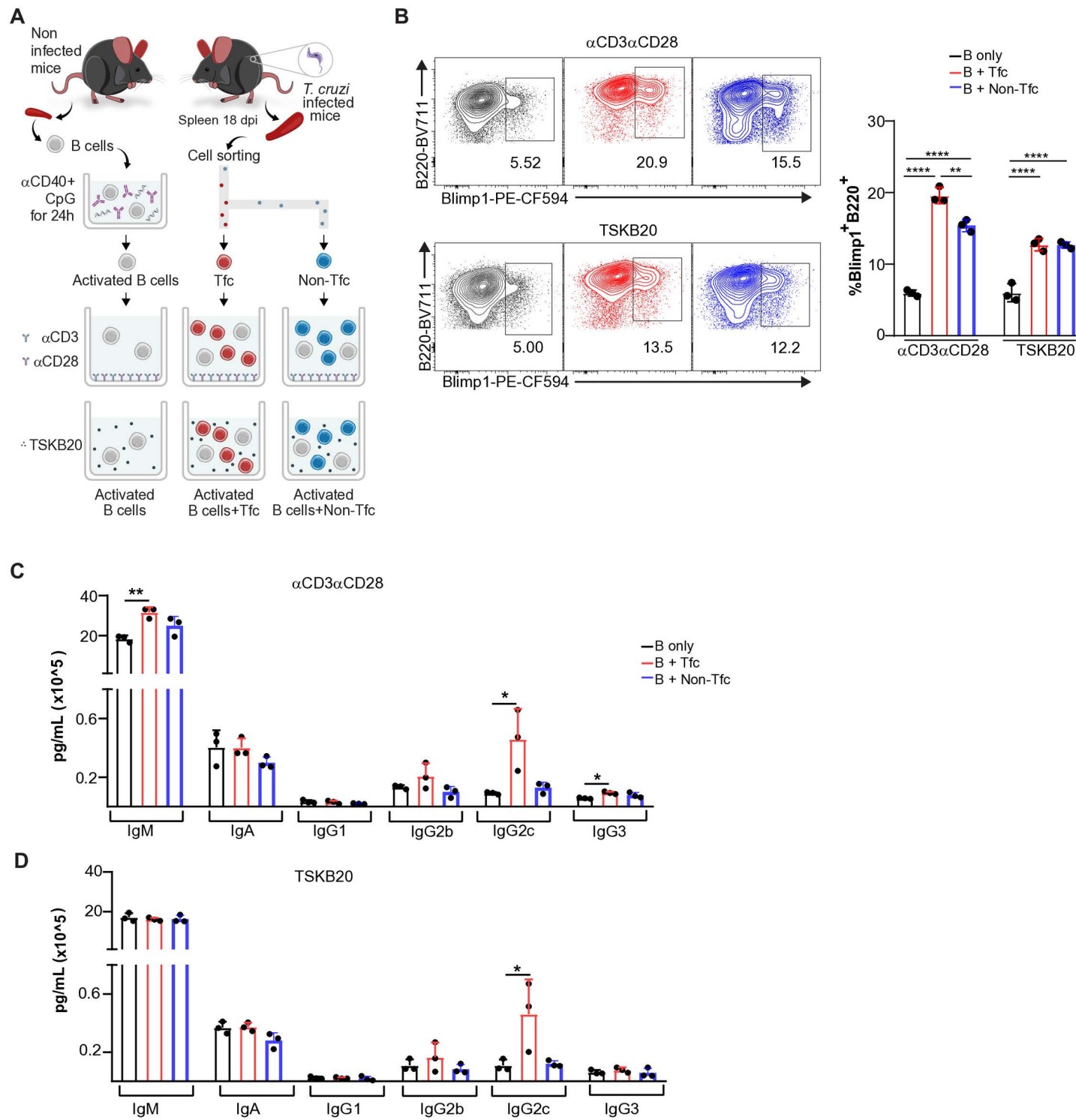

**Fig 6. Ab production and differentiation of activated B cells by Tfc and Non-Tfc cells.** B cells from non-infected mice were purified by immunomagnetic negative selection and stimulated with CpG plus anti-CD40. After 24 h, stimuli-activated B cells were washed and co-cultured with sorted Tfc and Non-Tfc cells. T cells were activated with anti-CD3 plus anti-CD28 or the TSKB20 peptide. **(A)** Schematic representation of the *in vitro* co-culture assay. **(B)** Representative contour plots (left panel) and statistical analysis of the percentage of Blimp-1⁺B220⁺ cells (right panel), gated on live CD8⁻B220⁺ cells. **(C-D)** Statistical analysis of Ig concentrations (IgM, IgA, IgG1, IgG2b, IgG2c, and IgG3) in culture supernatants in which T cells were

activated with (C) anti-CD3 plus anti-CD28 or **(D)** TSKB20. Each symbol represents an individual mouse. **(A-D)** Data are representative of 3 independent experiments. Statistical analysis: ordinary one-way ANOVA with selected comparisons and Bonferroni correction. *p<0.05, **p<0.01, ****p<0.0001.

in co-cultures with Tfc cells. In contrast, IgG1 and IgG3 levels remained unchanged in the presence of Tfh cells (S5C Fig). Both Tfc and Tfh cells reduced IgG2c levels, although the reduction was less pronounced in co-cultures with Tfh cells.

## Discussion

This study, in the context of *T. cruzi* infection, demonstrates that the CD8+T cell population characterized by high expression of CXCR5 and PD-1 corresponds to Tfc cells. We confirmed their identity through transcriptomic analysis and by detecting key proteins, including the transcription factors Bcl6 and TCF-1, as well as helper molecules such as IL-21, ICOS, and CD40L. We also identified CD8+cells within B lymphoid follicles in the spleens of *T. cruzi*-infected mice, coinciding with the peak of the Tfc cell response detected by flow cytometry. The high CXCR5 expression in B cells, compared to its expression in Tfc cells, limited the ability to reliably identify Tfc by immunofluorescence techniques on tissue sections. However, the spatial distribution of CD8+cells observed by immunofluorescence, together with flow cytometry data, supports the interpretation that the CD8+cells present in follicles likely correspond to the Tfc population.

Notably, CD8+T cells were observed within B cell follicles even before GC formation, suggesting that any interaction or effect of Tfc cells on B cells may occur before their differentiation into GC B cells. Unlike in other infections, where Tfc cells are mainly described during chronic phases [16–18], in *T. cruzi* infection we found that Tfc cells appear early, displaying kinetics that parallel parasitemia and the plasmablast response. This raises the possibility that parasitemia could drive the induction of both Tfc cells and plasmablasts, which secrete Abs capable of controlling the parasite [4]. Thus, Tfc cells may, directly or indirectly, influence parasite replication, although further studies are needed to confirm this.

Transcriptomic and flow cytometry analyses demonstrated that CXCR5+PD-1+CD8+T cells had a distinct phenotype from CXCR5-PD-1-CD8+T cells. We chose to perform bulk RNAseq on sorted populations in order to directly compare the transcriptional profiles of Tfc and Non-Tfc cells at the population level, in line with the objectives of this study. Although single-cell RNAseq of antigen-specific Tfc cells could provide complementary information regarding heterogeneity and functional states, bulk RNAseq proved to be a robust and informative approach for defining the core transcriptional program of Tfc cells during *T. cruzi* infection. Tfc cells expressed high levels of transcripts encoding molecules typically associated with Tfh cells, which were validated at the protein level. While transcriptomic analysis revealed similar levels of *Cxcr3* and *Cxcr4* transcripts in both Tfc and Non-Tfc cells, protein expression of both receptors was significantly higher in Tfc cells, as determined by flow cytometry. These results are consistent with previous findings indicating that CXCR3 expression is elevated in Tfc cells compared to conventional cytotoxic CD8+T cells [15,44], and suggest that CXCR3 may facilitate Tfc migration to inflamed tissues, such as the spleen, where infected B cell follicles are present. While CXCR4, as well as CXCR5, are required for proper GC location within the follicles [45], CXCR3 could also contribute to this process. Tfc cells showed lower *Ccr7* transcript levels than Non-Tfc cells, yet CCR7 protein expression remained higher in Tfc cells. This apparent discrepancy could potentially be explained by post transcriptional regulation mechanisms [46]. CCR7 is crucial for T cell migration toward the T cell zone in response to CCL19 and CCL21 [47]. In LCMV-infected mice, CXCR5+Tfc cells expressing high *Ccr7* reside in the T cell zone [48], whereas downregulation of CCR7 facilitates migration from the T cell zone or T–B border into the B cell area [47]. In our model, a subset of Tfc cells expressed CCR7, suggesting that those interacting with plasmablasts outside follicles may be CCR7+. This is in agreement with LCMV studies where CXCR5+CCR7+CD8+T cells are excluded from GCs [48].

Transcriptomic analysis further revealed that Tfc cells expressed higher levels of *Ccl3*, *Ccl4*, and *Xcl1*. These chemokines are associated with inflammatory responses: CCL3 promotes pro-inflammatory cytokine production and macrophage phagocytosis [49–51], CCL4 recruits immune cells to inflammatory sites [52], and XCL1 is chemotactic for B lymphocytes

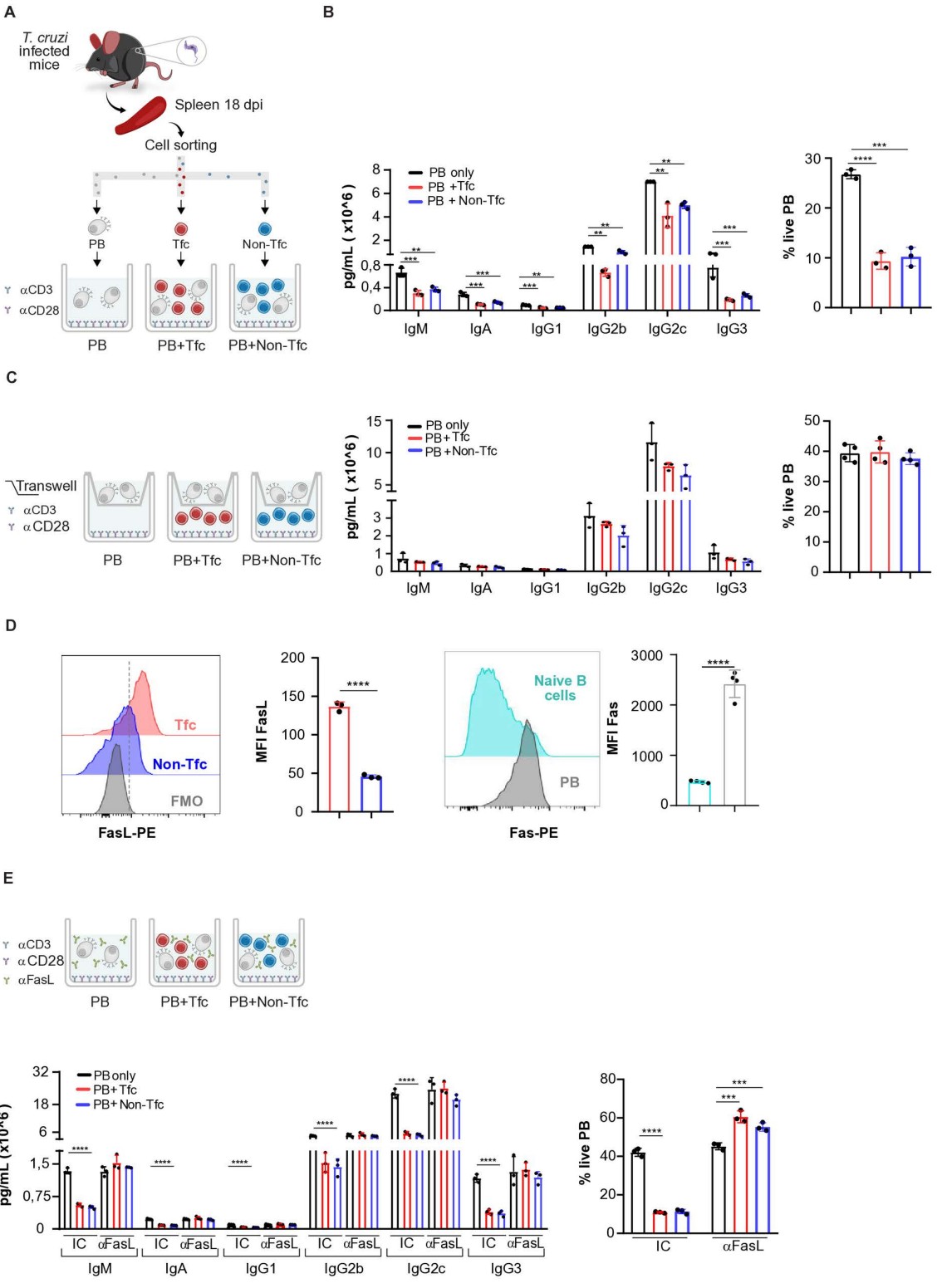

**Fig 7. Viability and Ab production by plasmablasts co-cultured with Tfc and Non-Tfc cells.** Plasmablasts (PB), Tfc, and Non-Tfc cells were sorted from the spleens of *T. cruzi*-infected mice at 18 dpi and co-cultured for 20 h in the presence of anti-CD3 plus anti-CD28. **(A)** Schematic representation of the *in vitro* culture assays. The frequency of live PB (right panel, gated on Caspase-3⁻ SYTOX⁻CD8⁻ B220ⁱⁿᵗ) and Ig concentrations (left panel) were

measured under different experimental conditions **(B, C, E)**. Co-cultures were performed: **(B)** without a transwell insert, **(C)** with a transwell, and **(E)** in the presence of anti-FasL blocking Ab (αFasL) or an isotype control Ab (IC). Additionally, **(D)** the MFI of FasL was measured on Tfc and Non-Tfc cells, while Fas expression was assessed in naïve B cells and in PB. Each symbol represents an individual mouse. Results are representative of two to three independent experiments. Statistical analysis: Ordinary one-way ANOVA with selected comparisons and Bonferroni correction. **p < 0.01, ***p < 0.001, ****p < 0.0001.

[53]. Together with co-stimulatory molecules, these chemokines likely contribute to the functional roles of Tfc cells in *T. cruzi* infection.

Contrary to previous reports suggesting a central memory phenotype for Tfc cells due to high CD62L expression [15], we found that most Tfc cells displayed an effector or effector memory phenotype (CD44$^+$CD62L$^-$). However, the presence of CCR7$^+$Tfc cells suggests that a minor central memory-like subset may also be present. Central memory T cells co-express CCR7 and CD62L and are able to migrate to secondary lymphoid organs in response to CCL19 and CCL21 [54,55]. Additionally, Tfc cells expressed the transcription factors T-bet and Eomes, which are involved in cytotoxicity and IL-2/PD-1 regulation [56–59]. This transcriptional profile is consistent with the higher frequency of IFN-γ$^+$ cells observed within the Tfc population. Also, Tfc expressed higher levels of TCF-1, as evidenced by MFI determined by flow cytometry. TCF-1, encoded by *Tcf7*, is a key transcription factor involved in T cell development and in the differentiation and function programming of mature CD8$^+$T cells, including the generation of memory subsets [30]. The elevated expression of TCF-1 observed in Tfc cells is consistent with the phenotype of this population detected in *T. cruzi* infected-mice, which includes central memory cells [30]. In addition, TCF-1, along with Bcl6, appears to contribute to the transcriptional program underlying Tfc differentiation [44].

It is interesting to note that CCR7 and TCF-1 expression was reduced in Non-Tfc cells from infected mice compared with naïve CD8$^+$T cells from non-infected mice, even though Non-Tfc cells still include some naïve CD8$^+$T cells. This reduction likely reflects infection-induced modulation of chemokine receptors, which may result from prolonged exposure to their ligands, potentially increased due to pathogen-driven inflammation, or could represent a pathogen strategy to prevent the migration of cells involved in host defense [60–62].

Tfc cells may represent a distinct and functional CD8$^+$T cell population that emerges during *T. cruzi* infection, characterized by features not merely attributable to an activated state. Functionally, Tfc cells from the spleens of *T. cruzi*-infected mice appears to be a polyfunctional subset [11], capable of producing various cytokines and effector molecules upon stimulation with either polyclonal or *T. cruzi*-specific stimuli. These include IL-4, IL-6, IL-21, Granzyme B, and Perforin. The frequency of Tfc cells expressing Granzyme B and Perforin is comparable to that observed in Non-Tfc cells (Fig 4C), suggesting that both subsets may exert cytotoxicity. Notably, Tfc cells from *T. cruzi* infected mice also produced IL-17A in significantly higher amounts than Non-Tfc cells, a finding not previously reported and whose functional implications in this context remain to be elucidated. Consistent with the transcriptional enrichment of *Il21*, flow cytometry revealed a higher proportion of IL-21$^+$cells among Tfc than among Non-Tfc cells. IL-21 plays a critical role in B cell help and plasma cell differentiation [63], suggesting that Tfc cells in *T. cruzi* infection could contribute to the establishment of early humoral immunity. *In vitro* co-culture experiments showed that naïve B cells incubated with Tfc cells displayed increased Blimp1 expression and secreted multiple Igs isotypes. Although Blimp1 is not strictly required to initiate Ab secretion, its expression is crucial for plasma cell maturation [64], and its induction supports the idea that Tfc cells can promote the differentiation of naïve B cells into Ab-secreting cells. Interestingly, Tfc cells induced Ab production even in the absence of direct contact, though optimal B cell activation required cell–cell interaction. In addition to naïve B cells, Tfc cells also favored Ab production in previously *in vitro* activated B cells, reinforcing their capacity to support different stages of B cell responses. Furthermore, Tfc cells induced class switching, predominantly toward IgG2c, which aligns with their high IFN-γ production and supports their role in shaping Th1-skewed humoral responses during *T. cruzi* infection. Consistent with our findings, Tyllis et al. [65] reported that CXCR5$^+$CD8$^+$T cells can promote IgG2c responses *in vivo*, further highlighting the potential role of Tfc cells in regulating isotype switching and shaping humoral immunity. When comparing the helper capacity of Tfc and Tfh cells, we found

that Tfh cells generally promoted higher Ab levels, although the differences were modest and isotype-dependent (S5 Fig). In line with this, no significant differences were detected for certain isotypes, including IgA, IgG2b, and IgG3. These findings suggest that the helper function of Tfc cells may be largely comparable to that of Tfh cells.

In parallel, Tfc cells also exhibited regulatory cytotoxic functions. Studies in LCMV and HIV infection have reported that Tfc cells lacking IL-21 but displaying potent cytotoxic activity eliminate infected Tfh cells within GCs [15,66]. In *T. cruzi*-infected mice, the presence of CXCR5+PD-1+CD8+cells in follicles could be driven by antigen concentration or the local inflammatory milieu [20,66,67]. We observed that both Tfc and Non-Tfc cells reduced Igs concentration and plasmablast viability through a Fas/FasL-dependent mechanism when they were co-cultured. Tfc cells expressed higher levels of FasL than Non-Tfc cells, and plasmablasts (short-lived extrafollicular cells) [6,7] expressed high levels of Fas. In contrast, naïve B cells express low Fas and are therefore less sensitive to FasL-induced apoptosis [68]. The use of a FasL-blocking Ab abrogated plasmablast death and restored Ig levels in co-cultures, indicating that CD8+T cells actively regulate the survival of Ab-secreting cells. Although CD8+T cells have been shown to suppress alloantibody responses by eliminating alloprimed IgG1+B cells [69], our results provide the first evidence of direct modulation of plasmablast viability by CD8+T cells during *T. cruzi* infection, a function mediated by Tfc cells and also detectable in Non-Tfc cells.

Our group has previously demonstrated that CD4+Tfh cells expressing Bcl6 are essential for the generation of extrafollicular plasmablast during *T. cruzi* infection [7]. In the current study, we confirmed that Tfh cells, as expected, promoted Ab production by naïve B cells *in vitro*, even under short-term co-culture conditions (20 h), and to a similar extent as CD8+Tfc cells, except for IgG2c and IgM. However, when co-cultured with differentiated B cells such as plasmablasts, Tfh cells reduced Ab production less efficiently than Tfc cells. Together, these findings support the idea that while CD4+Tfh cells are critical for the initiation of the plasmablast response, CD8+Tfc cells may act at a later stage to eliminate differentiated plasmablasts. Thus, Tfc-mediated killing could provide an additional mechanism of regulation to restrain potentially harmful humoral responses during infection. CD8+Non-Tfc cells were also able to control plasmablast viability. This suggests that both Tfc and Non-Tfc populations might contribute to the control of extrafollicular B cell responses, particularly targeting plasmablasts that have not undergone the same stringent selection mechanisms as GC B cells [70,71].

A dual role for Tfc cells, acting as cytotoxic effectors in infections and as B cell helpers in tumors or autoimmune settings, has been discussed in the literature [72]. Our data support this dual functionality showing that in *T. cruzi* infection Tfc cells not only promote B cell activation and Ab secretion but also regulate the response by eliminating plasmablasts. Their localization at the T–B border and within B cell follicles suggests a capacity to interact with different B cell subsets across maturation stages. Although Tfc cells may eliminate some parasite-specific Ab-secreting cells, this may help prevent the expansion of autoreactive clones. Moreover, their activity is concentrated in the acute phase of the infection, prior to the formation of GC-derived plasma cells responsible for secreting high-affinity Abs. Altogether, our findings position Tfc cells as key regulators of the humoral response in *T. cruzi*-infected mice, integrating effector, helper, and regulatory programs to balance protective immunity and immune control.

## Materials and methods

### Ethics statement

All animal experiments were approved and conducted in accordance with the guidelines of the Institutional Animal Care and Use Committee of the Facultad de Ciencias Químicas, Universidad Nacional de Córdoba (protocol numbers RD-723–2022 and RD-349–2022.).

### Mice and experimental infection

Wild-type C57BL/6 mice, aged 8–12 weeks and sex-matched, were used in all experiments. Mice were maintained under a 12 h light/dark cycle in temperature- and humidity-controlled conditions. This colony, originally obtained from the

Facultad de Veterinaria, Universidad Nacional de La Plata (La Plata, Argentina), has been bred in the Animal Facility of CIBICI-CONICET, FCQ-UNC for at least six years.

To induce *T. cruzi* infection, mice were intraperitoneally injected with $5 \times 10^3$ bloodstream trypomastigotes of the Tulahuén strain in 0.2 mL PBS. Non-infected littermates received 0.2 mL of PBS and were processed in parallel. Spleens were collected at various dpi for immune response analysis.

## Parasite quantification in blood

Parasitemia was assessed by counting viable bloodstream trypomastigotes in whole blood lysed with 0.87% ammonium chloride buffer, as previously described [4,73].

## Cell preparation

Spleens from infected and control mice were harvested and mechanically dissociated through 70 µm cell strainers. Red blood cells were lysed using Tris–ammonium chloride buffer for 5 min. Viable leukocytes were counted by trypan blue exclusion using a hemocytometer.

## Immunofluorescence

Spleens were collected and frozen in liquid nitrogen. Frozen sections (7 µm) were cut, fixed in cold acetone for 10 min, air-dried at room temperature, and stored at -80°C until use. Slides were hydrated in Tris buffer and blocked for 30 min at 25°C with 10% normal mouse serum–Tris buffer [74]. After blocking, slides were incubated for 50 min at 25°C with different combinations of the following anti-mouse Abs: anti-B220 (Cat# RM2620, clone RA3-6B2, dilution 1/100) from Life Technologies; PE-labeled anti-CD8a (Cat# 12-0081-83, clone 53-6.7, dilution 1/150), and Alexa Fluor 647-labeled-Lectin PNA from eBiosciences (Cat# L21409, dilution 1/200); PE (Cat# 553714) and BV421(Cat# 562610)-labeled anti CD138 (clone 281–2, dilution 1/100), and BV421-B220 (Cat# 562922, clone RA3-6B2, dilution 1/100) from BD Biosciences. Tissue sections were mounted with FluorSave (Merck Millipore) and images were acquired using an Olympus FV1200 confocal microscope and analyzed with ImageJ64 v1.52e (National Institutes of Health, USA).

## Abs and flow cytometry

For surface staining, cell suspensions were incubated with fluorochrome-labeled Abs along with Live/Dead Fixable Aqua 405 (Invitrogen, Cat# L34966, dilution 1/400) or NIR Fixable Viability Kit 633 (Invitrogen, Cat# L10119, dilution 1/800) in ice-cold PBS 2% FBS for 25 min at 4°C. Different combinations of anti-mouse Abs were used (S1 Table). *T. cruzi*-specific CD8$^+$ T cells were evaluated using a BV241 or APC-labeled tetramer of H-2K(b) molecules loaded with *T. cruzi* trans-sialidase immunodominant ANYKFTLV (Tskb20) peptide (NIH Tetramer Core Facility). CD40L expression was assessed on the cell surface. CCR7 expression was first measured at 37 °C for 30 min, after which the remaining markers were stained. After staining, cells were washed and acquired using a LSRFortessa X-20 flow cytometer (BD Biosciences). To identify CXCR5$^+$ cells, cell suspensions were labeled with Biotin-CXCR5 (BD Biosciences, Cat# 551960, clone 2G8, dilution 1/75) for 30 min at room temperature, followed by surface staining with streptavidin-PE (eBioscience, Cat# 12-4317-87, dilution 1/300), APC (Biolegend, Cat# 554067, dilution 1/300), or PECy7 (eBioscience, Cat# 25-4317-82, dilution 1/300) along with other Abs. Transcription factors were detected following cell fixation and permeabilization using the Foxp3 Staining Buffer Set, according to the manufacturer's protocol (Thermo Fisher Scientific). Abs also were listed in S1 Table.

Data were analyzed using FlowJo software (versions X.0.7 and 10.8.1), with gating coordinates established using negative controls for population markers. FMO controls were used for activation markers, regulatory molecules, or continuously expressed molecules.

## CD8$^+$ T cell effector function assay

Spleen cell suspensions were cultured for 5 h in RPMI 1640 medium (Gibco) supplemented with 10% heat-inactivated FBS (Gibco), 2 mM glutamine (Gibco), 55 µM 2-ME (Gibco), and 40 µg/mL gentamicin. Cells were either cultured with medium alone or stimulated with 5 µg/mL TSKB20 (ANYKFTLV) peptide (Genscript Inc.), or PMA (Sigma-Aldrich, Cat# P1585), and 1 µg/mL ionomycin (Sigma-Aldrich, Cat# I0634) in the presence of Brefeldin A and Monensin (eBioscience, Cat# 00-4506-51 and Cat# 00-4505-51). Anti-CD107a was included during the culture period. Following incubation, cells were surface-stained, fixed, and permeabilized using BD Cytofix/Cytoperm and Perm/Wash solutions (BD Biosciences). Intracellular staining was performed with anti-mouse Abs listed in S1 Table. Stained cells were acquired on a LSRFortessa X-20 flow cytometer (BD Biosciences) and analyzed using FlowJo software.

## Glucose uptake assay

Splenocytes from 18 dpi-infected and control mice were incubated in low-glucose medium (DMEM with low glucose) at 37°C and 5% $CO_2$ for 30 min. The fluorescent glucose analog 2-NBDG (2-(N-(7-nitrobenz-2-oxa-1,3-diazol-4-yl) amino)-2-deoxyglucose)), which competes with D-glucose for cell entry via GLUT1 transporters, was then added for an additional 30 min [35]. Later, surface markers were stained as described in "Abs and Flow Cytometry".

## Cell sorting

Spleen cell suspensions were stained with fluorochrome-labeled anti-mouse Abs CD8- PE-Cy7, CD4-FITC, B220-FITC, CXCR5-Biotin, PD-1-PECy7 and APC, IgD-FITC, CD138-APC, Streptavidin-PE, IgM-PECy7, and LIVE/DEAD cell. Cells were washed in PBS containing EDTA and 0.4% FBS and were sorted in BD FACSAria II cytometer (BD Biosciences). Tfc (CXCR5$^+$PD-1$^+$CD4$^-$B220$^-$CD8$^+$), Non-Tfc (CXCR5$^-$PD-1$^-$CD4$^-$B220$^-$CD8$^+$), naïve B (IgD$^+$IgM$^+$B220$^+$), plasmablasts (IgD$^-$CD138$^+$B220$^{int}$), and Tfh (CXCR5$^+$PD-1$^+$CD8$^-$B220$^-$CD4$^+$) cells were sorted from live lymphocytes. Naïve B cells were also sorted from uninfected mice.

## B cell activation

Splenic B cells from non-infected mice were isolated using a negative selection kit (StemCell, Cat#19854) according to the manufacturer's instructions. A total of $2 \times 10^5$ B cells were cultured in complete RPMI with 2 µg/mL CpG (Eurofins Genomics) and 2 µg/mL anti-CD40 (BioLegend) for 24 h at 37°C in a 96-well plate. Throughout the manuscript, these cells are referred to as stimuli-activated B cells. Following stimulation, B cells were washed with complete RPMI, counted, and subsequently used in co-culture assays.

## *In vitro* co-cultures

For co-culture experiments, $5 \times 10^4$ sorted naïve B cells (IgD$^+$IgM$^+$B220$^+$), plasmablasts (IgD$^-$CD138$^+$B220$^{int}$), or stimuli-activated B cells were co-cultured with $1 \times 10^5$ sorted Tfc (CXCR5$^+$PD-1$^+$CD8$^+$) or Non-Tfc (CXCR5$^-$PD-1$^-$CD8$^+$) cells in 96-well U-bottom plates pre-coated with anti-CD3 (Thermo Fisher Scientific, Cat# 14-0031-85, 0.5 µg/mL) plus anti-CD28 (Thermo Fisher Scientific, Cat# 14-0281-86, 0.2 µg/mL), or with TSKB20 peptide. Cultures were maintained in complete RPMI for 20 h at 37 °C.

For co-cultures involving Tfh cells, $5 \times 10^4$ naïve B cells or plasmablasts were co-cultured with $1 \times 10^5$ sorted Tfh cells (CXCR5$^+$PD-1$^+$CD8$^-$B220$^-$CD4$^+$) under the same stimulation conditions (anti-CD3 plus anti-CD28, 0.5 µg/mL and 0.2 µg/mL, respectively) and incubation time.

Transwell assays were performed in 96-well plates similarly coated with anti-CD3 plus anti-CD28. $1 \times 10^5$ Tfc or Non-Tfc cells were seeded in the lower chamber, and $5 \times 10^4$ naïve B cells or plasmablasts were added to the transwell inserts.

Blocking experiments were carried out using the same co-culture setup in the presence of anti-mouse Fas Ligand Ab (BioXcell, Cat# BE0319, 5 µg/mL) or anti-IFN-γ Ab (BioLegend, Cat# 505834, 5 µg/mL). Corresponding isotype control

Abs were used: Armenian Hamster IgG (BioLegend, Cat# 400940, 5 µg/mL) and Purified Rat IgG1, κ (BioLegend, Cat# 400431, 5 µg/mL), respectively.

After 20 h of culture, Ig isotype concentrations were measured in the supernatants. Cells were collected and stained for flow cytometry using fluorochrome-conjugated anti-mouse Abs (listed in S1 Table), including IgD-BV786, MHC-II-Super Bright 600, B220-BV711, CD8-PECy7, Fas-PE, and Blimp1-PECF594.

To evaluate plasmablast survival in co-cultures with Tfc, Non-Tfc or Tfh cells, we used the CellEvent Caspase-3/7 Detection Kit (Invitrogen, Cat# C10423). Live plasmablasts were identified as Caspase-3$^-$ SYTOX$^-$CD8$^-$ B220$^{int}$. Data acquisition was performed using a LSRFortessa X-20 flow cytometer.

### Immunoglobulin quantification

The concentrations of Ig isotypes (IgM, IgG1, IgG2b, IgG2c, IgG3, and IgA) in the supernatants from the various co-culture experiments were quantified using a bead-based assay (BioLegend, Mouse Immunoglobulin Isotyping Panel, 6 plex, Cat# 740493) and analyzed by flow cytometry. Samples and standards were run in duplicate according to the manufacturer's instructions. The samples were transferred from plate to tube and read using a FACSCanto II cytometer and data was analyzed with Data Analysis Software (Biolegend). For statistical analysis, samples with non-detectable (ND) values were imputed with a value equal to the lower detection limit of the technique employed.

### Cytokines quantification

Tfc and Non-Tfc cells were sorted from spleens of *T. cruzi*-infected mice at 18 dpi, as previously described, and stimulated for 4 h with anti-CD3 plus anti-CD8 or with PMA/ionomycin. Cytokine concentrations (IFN-γ, TNF, IL-2, IL-10, IL-17A, and IL-6) in cell supernatants were measured using a bead-based assay Mouse Th Cytokine and Mouse Inflammation (Biolegend, Cat# 741043 and Cat# 740150 respectively) panels, according to the manufacturer's instructions.

### RNA sequencing and analysis

Tfc and Non-Tfc cells were sorted from spleens of *T. cruzi*-infected mice at 18 dpi, as previously described. Total RNA was extracted with the Arcturus PicoPure RNA Isolation Kit (Thermo Fisher Scientific), as indicated by the manufacturer. RNA sequencing (RNA-Seq) was performed by RNA-Seq services by GENEWIZ Multiomics & Synthesis Solutions from Azenta Life Sciences (NJ, USA). PolyA selection for mRNA species was used for rRNA removal and GENEWIZ performed an ultra-low input library preparation.

For the analysis, Sequence reads were trimmed to remove possible adapter sequences and nucleotides with poor quality using Trimmomatic v.0.36. The trimmed reads were mapped to the Mus musculus GRCm38 reference genome available on ENSEMBL using the STAR aligner v.2.5.2b. Using DESeq2, a comparison of gene expression between the customer-defined groups of samples was performed. The Wald test was used to generate p-values and log2 fold changes. Genes with an adjusted p-value < 0.05 and absolute log2 fold change > 1 were called differentially expressed genes for each comparison. The datasets generated for this study can be found in the NIH repository under accession number PRJNA1234210 (https://www.ncbi.nlm.nih.gov/sra/PRJNA1234210).

### Statistical analysis and data visualization

Unless otherwise indicated, statistical analyses and graph generation were performed using GraphPad Prism version 8.0.1. Data normality was assessed using the Shapiro-Wilk test. For normally distributed data, comparisons between groups were made using Student's *t*-test or one-way ANOVA followed by Bonferroni's post-hoc test. For non-normally distributed data, the Mann-Whitney *U* test or Kruskal-Wallis test was used, as appropriate. P values ≤ 0.05 were considered statistically significant and are indicated in the figures. Outliers were identified using the ROUT method. Data are

presented as mean ± standard deviation (SD), and the number of animals is indicated in the figure legends or shown in the plots.

Flow cytometry data were analyzed using FlowJo software version X.0.7. Visualizations such as heatmaps, volcano plots, and principal component analysis (PCA) were generated using the SRplot online platform [75]. Gene Set Enrichment Analyses (GSEA) was conducted to identify enriched pathways of whole transcriptomics of Tfc cells. This analysis was performed using the GSEA software and gene sets from the Molecular Signatures Database (MSigDB), specifically focusing on the M5 mouse collection [76].

### AI language model assistance

We used ChatGPT (developed by OpenAI) to assist in refining the written content of this study. ChatGPT provided corrections based on the input provided by the user, enhancing the clarity and grammar of the text. ChatGPT output was critically revised by the user to ensure it conveys the desired message.

### Supporting information

**S1 Fig. Gating strategy in flow cytometry.** Representative flow cytometry contour plots illustrating gating strategies. After exclusion of doublets, lymphocytes were identified based on forward (FSC-A) and side scatter (SSC-A) parameters. Live cells were gated by excluding those stained with Live/Dead Fixable Aqua 405. (A) B cells were selected based on B220 expression. Within this population, GC B cells were identified based on Fas and GL-7 expression, while plasmablasts (PB) were identified within live lymphocytes based on B220$^{int}$ and CD138 expression. (B) Representative flow cytometry plots showing the gating strategy used to identify Tfc CD8$^+$T cells (CXCR5$^+$PD-1$^+$) and Non-Tfc (CXCR5$^-$PD-1$^-$) CD8$^+$T cells from the spleens of *T. cruzi*-infected mice. Gating was performed sequentially on singlets, live CD8$^+$cells (CD4$^-$B220$^-$), and subsequently on CXCR5 vs. PD-1 expression. Percentages represent the proportion of cells within the respective parent gate. (C) Representative flow cytometry plots illustrating the gating strategy used to identify Tfh CD4$^+$T cells (CXCR5$^+$PD-1$^+$) from the spleens of *T. cruzi*-infected mice. Gating was performed sequentially on singlets, live CD4$^+$cells (CD8$^-$B220$^-$), followed by CXCR5 and PD-1 expression. Percentages indicate the proportion of cells within the respective parent gate. (D) Assessment of cytokine expression (IFN-γ, TNF, IL-6, IL-21, IL-10, IL-4), GrzB, Prf, and CD107a in Tfc (CXCR5$^+$PD-1$^+$) and Non-Tfc (CXCR5$^-$PD-1$^-$) CD8$^+$ T cells following stimulation with either PMA/ionomycin or the specific peptide TSKB20. Each plot shows the frequency of Tfc or Non-Tfc CD8$^+$T cells expressing the indicated effector molecule. Percentages within the quadrants indicate the proportion of cytokine-positive cells within the respective gated population.
(TIF)

**S2 Fig. Gene ontology analysis.** Gene Ontology (GO) enrichment analysis based on RNA-seq data from Tfc cells, respect to Non-Tfc cells. The plot displays up to 40 significantly enriched GO terms (adjusted p-value, -log10(adjusted p-value)), focusing on categories related to immune responses, cellular activation, and cytokine regulation.
(TIF)

**S3 Fig. Expression of B cell-help molecules, chemokine receptors and transcription factors in Tfc, Non-Tfc and naïve CD8$^+$ T cells.** Splenic cells from non-infected and *T. cruzi* infected mice were collected at 18 dpi and analyzed by flow cytometry. (A) Representative contour plots showing the gating strategy used to identify CD44$^+$Non-Tfc (CXCR5$^-$PD-1$^-$CD8$^+$) T cells. (B–D) Representative histograms and corresponding quantification of MFI for (B) B cell help-associated molecules (ICOS, CD40L, and Bcl6), (C) chemokine receptors (CXCR3, CXCR4, and CCR7), and (D) transcription factors (T-bet, Eomes, and TCF-1) in Tfc cells (all CD44$^+$) and activated (CD44$^+$) Non-Tfc CD8$^+$T cells. (E) Representative histograms and quantification of MFI and frequency of CCR7$^+$ and TCF-1$^+$cells determined using FMO

controls for Tfc and Non-Tfc cells (without excluding naïve T cells) from infected mice, and for naïve CD8$^+$T cells from non-infected mice (CD8$^+$naïve NI, light blue). Data are presented as mean ± SD; N = 3 (A-D) and N = 4 mice. Data were collected from 2 independent experiments. Statistical significance was determined by an unpaired t-test (A-D) and an unpaired t-test by comparing CD8$^+$ naïve NI - Tfc and CD8$^+$ naïve NI - Non-Tfc cells (*p < 0.05, **p < 0.01, ***p < 0.001, ****p < 0.0001).
(TIF)

**S4 Fig. Ab secretion by naïve B cells from non-infected mice.** Naïve B cells from non-infected mice were co-cultured with either Tfc or Non-Tfc cells isolated from infected mice at 18 dpi. The graph shows the concentrations of Igs measured in the culture supernatants. ND: not detected. For statistical purposes the samples with ND values were imputed with a value equal to the lower detection limit of the technique employed. Data are presented as mean ± SD. Each symbol indicates an individual animal, N = 3. Data are representative of two independent experiments. Statistics: ordinary one-way ANOVA with select comparisons and a Bonferroni correction. *p < 0.05, ***p < 0.001, ****p < 0.0001.
(TIF)

**S5 Fig. Ab production by naïve B cells and plasmablasts in the presence of Tfc and Tfh cells.** (A) Sorted naïve B cells or (C) plasmablasts were co-cultured for 20 h with either medium alone (black), sorted Tfc cells (red), or sorted Tfh cells (green), in the presence of anti-CD3 plus anti-CD28. All cell populations were purified from the spleens of *T. cruzi*-infected mice at 18 dpi. (A, C) Statistical analysis of Ig concentrations (IgM, IgA, IgG1, IgG2b, IgG2c, and IgG3) measured in the culture supernatants using a multiplex bead-based assay. (B) Representative histogram and statistical analysis of MFI of FasL. Data are presented as mean ± SD. Each dot represents an individual mouse, N = 3. (A-C) Data are representative of 2 independent experiments. Statistical analyses: ordinary one-way ANOVA with selected comparisons and Bonferroni correction (A, C); unpaired *t*-test (B). *p < 0.05, **p < 0.01, ***p < 0.001, ****p < 0.0001.
(TIF)

**S1 Table. List of Abs used for flow cytometry.** Detailed list of Abs used in flow cytometry experiments, including their target specificity, fluorochrome conjugate, supplier, catalog number, clone, and working dilution.
(TIF)

## Acknowledgments

We thank the staff of the facilities at CIBICI: M. P. Abadie, P. M. Crespo, S. Boccardo, V. Blanco, D. Lutti, C. Noriega, F. A. Frontera, S. R. Oms, R. E. Villarreal, G. Furlán, N. M. Maldonado, M. S. Miró, and Carlos Mas (CEMINCO) for their excellent technical assistance. We are also grateful to Cinthia Stempin (CIBICI) for their valuable scientific contributions and expertise with glucose uptake experiments. We acknowledge the NIH Tetramer Core Facility for provision of APC- and Brilliant Violet 421-labeled TSKB20 tetramers. CLM, EVAR and AG are Researchers from CONICET. YG, LA and JCG thank CONICET for the fellowship awarded. The content is solely the responsibility of the authors and does not necessarily represent the official views of the funding agencies.

## Author contributions

**Conceptualization:** Adriana Gruppi.

**Formal analysis:** Yamila Gazzoni.

**Funding acquisition:** Adriana Gruppi.

**Investigation:** Yamila Gazzoni, Laura Almada, Julio C Gareca.

**Methodology:** Yamila Gazzoni.

**Project administration:** Carolina L Montes, Eva V Acosta-Rodríguez, Adriana Gruppi.

**Resources:** Eva V Acosta-Rodríguez, Adriana Gruppi.

**Supervision:** Adriana Gruppi.

**Validation:** Yamila Gazzoni, Adriana Gruppi.

**Visualization:** Yamila Gazzoni, Adriana Gruppi.

**Writing – original draft:** Yamila Gazzoni, Eva V Acosta-Rodríguez, Adriana Gruppi.

**Writing – review & editing:** Yamila Gazzoni, Laura Almada, Julio C Gareca, Carolina L Montes, Eva V Acosta-Rodríguez, Adriana Gruppi.

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
