## [Decision Letter · Decision Letter 0]

26 May 2025

Follicular CD8+ T cells in Trypanosoma cruzi infection: helpers or killers depending on the target B cell population

PLOS Pathogens

Dear Dr. Gruppi,

Thank you for submitting your manuscript to PLOS Pathogens. After careful consideration, we feel that it has merit but does not fully meet PLOS Pathogens's publication criteria as it currently stands. Therefore, we invite you to submit a revised version of the manuscript that addresses the points raised during the review process.

Please submit your revised manuscript within 60 days Jul 25 2025 11:59PM. If you will need more time than this to complete your revisions, please reply to this message or contact the journal office at plospathogens@plos.org. Please include the following items when submitting your revised manuscript:

We look forward to receiving your revised manuscript.

Kind regards,

Martin Craig Taylor

Guest Editor

PLOS Pathogens

Dominique Soldati-Favre

Section Editor

PLOS Pathogens

Editor-in-Chief

PLOS Pathogens

orcid.org/0000-0003-2946-9497

Editor-in-Chief

PLOS Pathogens

orcid.org/0000-0002-7699-2064

**Additional Editor Comments :**

This manuscript has been reviewed by three experts in the field and they have raised concerns which need to be addressed before this manuscript can be considered further. Please respond to the reviewers comments, in particular those regarding image resolution, cell identification and contamination with naïve CD8+ T cells. The issues that need attention are detailed in the reviewers comments attached.

**Journal Requirements:**

At this stage, the following Authors/Authors require contributions: Yamila Gazzoni, Laura Almada, Julio Cesar Gareca, Carolina Lucia Montes, Eva Virginia Acosta-Rodriguez, and Adriana Gruppi. Please ensure that the full contributions of each author are acknowledged in the "Add/Edit/Remove Authors" section of our submission form.

- ® on page: 28

- TM on pages: 27, and 28.

Potential Copyright Issues:

i) Figures 5A, 5C, 5D, 5E, 6A, 7A, 7C, and 7E. Please confirm whether you drew the images / clip-art within the figure panels by hand. If you did not draw the images, please provide (a) a link to the source of the images or icons and their license / terms of use; or (b) written permission from the copyright holder to publish the images or icons under our CC BY 4.0 license. Alternatively, you may replace the images with open source alternatives. See these open source resources you may use to replace images / clip-art:

5) Thank you for stating "The datasets generated for this study can be found in the NIH repository under accession number PRJNA1234210 (https://www.ncbi.nlm.nih.gov/sra/PRJNA1234210).This link reaches a 404 error page. Please amend this to a new link. Please note that, though access restrictions are acceptable now, your entire data will need to be made freely accessible if your manuscript is accepted for publication. This policy applies to all data except where public deposition would breach compliance with the protocol approved by your research ethics board. If you are unable to adhere to our open data policy, please kindly revise your statement to explain your reasoning and we will seek the editor's input on an exemption. Please be assured that, once you have provided your new statement, the assessment of your exemption will not hold up the peer review process.

6) Please ensure that the funders and grant numbers match between the Financial Disclosure field and the Funding Information tab in your submission form. Note that the funders must be provided in the same order in both places as well. Currently, these grants "PICT 2018-01494" and "PIP 201511220150100560CO" are missing from the Funding Information tab.

7) Please provide a completed 'Competing Interests' statement, including any COIs declared by your co-authors. If you have no competing interests to declare, please state "The authors have declared that no competing interests exist". Otherwise please declare all competing interests beginning with the statement "I have read the journal's policy and the authors of this manuscript have the following competing interests:"

**Reviewers' Comments:**

Reviewer's Responses to Questions

**Part I - Summary**

Reviewer #1: In this study, the authors use mice infected with Trypanosoma cruzi, the causative agent of Chagas disease, to examine cytotoxic CD8+ T follicular helper cells (Tfc), and in particular, their roles in influencing parasite-specific B cell responses. They report that Tfc promote GC B cell responses, while inhibiting plasmablast antibody production via Fas/FasL interactions, although this latter finding wasn’t restricted to Tfc, but also occurred in the presence of non-Tfc.

Reviewer #2: The authors demonstrated that CD8+ T cells with very specific characteristics, which are found in or around germinal centers, are extremely important during T. cruzi infection. These cells are called Tfc (follicular cytotoxic T cells) and assist in B cell differentiation and antibody secretion. The methodology is appropriate for the study's objectives and ranges from transcriptome analyses throughout the infection to functional assays. Interestingly, the study also shows that these cells may have a regulatory function, destroying plasmablasts through the Fas-FasL pathway, and thus contributing to the homeostasis of the immune response.

Reviewer #3: In this manuscript, Yamila Gazzoni and colleagues perform transcriptomic and flow cytometric characterisation of CXCR5+PD-1+CD8+ T cells, referred to as T follicular cytotoxic (Tfc) cells, in a mouse model of Trypanosoma cruzi (T. cruzi) infection. Additionally, through a series of in vitro co-culture assays, the authors explore the helper and regulatory activities of Tfc cells in B cell responses, when compared to non-Tfc cells. While this work presents a novel step forward in characterising the biology of Tfc cells, there are issues that need to be addressed before publication can be considered.

**Part II – Major Issues: Key Experiments Required for Acceptance**

Reviewer #1: See also part III. The main issue to address is making the images shown in Figures 1D & E more convincing. As it stands, I cannot see discrete CD8+ T cells in the germinal centres or surrounding plasmablasts.

Reviewer #2: Although the authors have clearly demonstrated the functions of Tfc cells, it would be interesting, in a future experiment, to also investigate how Tfh cells assist B cells and whether the latter might have a regulatory role, given that they may also express Eomes, as shown for Tfc cells, and thus could potentially acquire this regulatory function.

Another important point concerns CD107, which is more highly expressed in Tfc cells compared to non-Tfc cells. This is an important marker of degranulation and effector function. However, it is not clear whether B cells themselves could also be infected and targeted for destruction by these cells, or if their function is restricted solely to plasmablasts.Although the authors have demonstrated the role of Tfc cells during the acute phase of T. cruzi infection, and their expansion appears to correlate with parasitemia, it remains unclear whether these cells exert a more cytotoxic function during the chronic phase of the infection.

Reviewer #3: Throughout the manuscript, the authors compare Tfc cells with non-Tfc cells, defined as CXCR5+PD-1+CD8+ and CXCR5-PD-1-CD8+ T cells, respectively. However, the gating strategies for these cell populations does not include any T cell activation markers (such as CD44), and as a result the non-Tfc population contains a significant proportion of naïve T cells (see Figure 4A). In fact, selecting PD-1- cells likely further enriches for naïve CD8+ T cells in the non-Tfc population. This issue complicates interpretations of phenotypic and functional differences between Tfc and non-Tfc cells, as the authors are essentially comparing an activated T cell population (Tfc) with a heterogenous population of T cells (non-Tfc) containing a large proportion of naïve, non-activated T cells, leading to some instances where the results are difficult to interpret. For instance, the RNA sequencing analysis indicates Tcf7 is expressed at higher levels in non-Tfc cells compared to Tfc cells, which contrasts with previously published results (PMIDs: 27487330, 27501248, 30377045). However, this result is likely due to the presence of a significant proportion of naïve CD8+ T cells (which highly express Tcf7) in the non-Tfc population. The authors should repeat analyses with the addition of T cell activation markers to identify both Tfc and non-Tfc cells. At the very least, the authors need to be more circumspect with their interpretations of comparative analyses between Tfc and non-Tfc cells to account for this issue.

**Part III – Minor Issues: Editorial and Data Presentation Modifications**

Reviewer #1: Supp 1B: This figure indicates that non-Tfc includes non-CD8+ T cells (eg., CD4, B cells macrophages, etc.). If this is the case, I don’t think it is appropriate, as the comparison should be with non-Tfc CD8+ T cells. If the latter is the case, please change the Figure to reflect this, as indicated in the text (line 163).

Figure 1D & E:

Better quality images are needed to show what is described in the text. For example, it is difficult to see CD8+ cells in the germinal centre (GC) at day 15 post-infection and determine if they are part of an extended peri-arteriol lymphoid sheath or represent distinct cells in the GC in Fig 1D.

Similarly, it is not clear there is co-localisation between CD8 and CD138 at day 18 p.i., in Fig 1E.

Note: I could only access a pdf version of the Figures.

It is not clear what the difference is between the upper and lower panels in Fig 1D at days 15, 18 and 23 p.i., and why two panels are shown for each time point.

Figure 2:

Show the flow cytometry plots to indicate how cells were sorted (these can be Supplementary files).

It is not clear why the authors refer to these cells as “cytotoxic” in the introduction, as the results in Fig 2C do not support this (i.e., IFNg and TNF up but granzymes down). This should at least be discussed.

Figure 3:

Show how CD4 Tfh cells were gated in Fig 3B.

Representative flow plots should be shown for Fig 3C & D.

Figure 4:

Show how Tfc and non-Tfc were gated in Fig 4A.

Show gating used for Tfc and non-Tfc in Fig 4D.

Figures 5-7:

Clear experimental outline and execution.

Discussion:

The authors should acknowledge limitations in their study, and in particular, short-comings in performing bulk RNAseq versus single cell RNAseq on antigen-specific Tfc cells (using available tetramer). In addition, I feel the authors gloss over the cytotoxic features of the Tfc in T. cruzi infection. Their data indicates they are not as cytotoxic as in HIV or LCMV. Additionally, the potential role of TCF-1 expression is not discussed.

Reviewer #2: The manuscript is very well written, with clearly defined objectives and a methodology that is appropriate and sufficiently rigorous to support the conclusions drawn. The authors employ a comprehensive experimental approach, ranging from transcriptomic analyses at different stages of infection to functional assays, which strengthens the validity of their findings. There are no major methodological flaws or ambiguous interpretations that would require clarification. The data are presented clearly and are well-supported by appropriate analyses

Reviewer #3: Minor Issues:

• For all data presented, the number of biological and/or technical replicates and the number of independent experiments performed for each analysis should be clearly indicated for each dataset. This is currently difficult to determine from some of the figure legends.

• It appears the data in Figure 5B are used again in Figure 5D. Were these experiments conducted separately, as indicated in the figure legend, or at the same time? If the data are from the same experiment this should be clearly communicated in the figure legend.

• The authors identify a role for Tfc cells in promoting IgG2c production by B cells in in vitro co-cultures. A previous publication looking at the role of CXCR5+CD8+ T cells in shaping antibody responses to immunisation and peripheral viral infection has demonstrated these cells promote IgG2c responses in vivo (PMID: 34326833). This study should be referenced in this manuscript.

• Line 149, the anatomical compartment (presumably blood?) in which parasitemia is assessed should be included in the figure legend. Additionally, the assay for determining parasitemia should be described in the methods.

• Line 157, the authors should be specific as to which data in the figure are from three independent experiments.

• Line 183, Mki67 is not a transcription factor encoding gene.

• Line 202, include time point (18-dpi?) of analysis in the figure legend.

• Line 223, how was CD40L measured, intracellular or surface staining? This should be detailed in the figure legend and methods section.

• Lines 276-278, a reference should be provided to support this conclusion.

• Line 317, IgM is not a class-switched Ig.

• The experiment in Figure 5E should be performed with an isotype control antibody in the control cultures.

• Line 359, what is the assigned value below the LOD? Is it zero?

• Lines 482-486, references should be provided to support each point in this sentence.

• Line 511, what is meant by ‘fully’ naïve B cells?

• Line 531, the conclusion that this reports the first evidence of direct modulation of plasmablast viability by Tfc cells requires more nuance as this capability is not unique to Tfc cells in T cruzi infection. As identified in this study, non-Tfc cells appear to be similarly able to kill plasmablasts ex vivo.

• All flow cytometry plots should be presented as pseudocolour dot plots or contoured plots with outliers, not as density plots.

• Axes for flow plots should also include fluorophore labels

• Given naïve cells are present in the non-Tfc population, what level of confidence do the authors have that the flow cytometry data accurately reflect protein expression of CCR7 and TCF-1 as presented in Figure 3 C and D. The flow cytometry appears to contradict the RNA sequencing results. The authors should comment on this.

• Line 532, the authors content that due to the lack of tolerance checkpoints for plasmablasts their elimination by Tfc cells may contribute to regulation of humoral immunity. However, this statement is not consistent with published results. In T cruzi infection, this group has previously shown that plasmablast responses in this model are dependent on a checkpoint involving BCL6-expressing CD4+ T cells (PMID: 35651611).

• The manuscript would benefit from a thorough grammatical review, especially the methods section and reference list.

• Line 565, presumably 70 μm (not 0.70 μm) cell strainers were used in cell preparation?

• The link to the RNA sequencing data set does not seem to work. This should be rectified before publication.

Suggestions and Questions

• In figure 4B and 4C, have the authors assessed viability of B cells in co-cultures where CD8+ T cells are stimulated with anti-CD3/anti-CD28 or TSKB20? If so, are there any difference in B cell viability between Tfc and non-Tfc co-cultures in the anti-CD3/anti-CD28 or TSKB20 stimulation conditions? This may be interesting as TSKB20 would likely be presented on MHC-I on B cells in this setting, leading to cognate interactions between CD8+ T cell populations and B cells.

• Have the authors assessed ICOSL expression on CXCR5+CD8+ T cells in T cruzi infection? This should be assessed considering ICOSL is expressed by regulatory CXCR5+CD8+ T cells in some contexts (PMID: 20844537).

• Do B cells with neutralising specificities become infected with T cruzi, resulting in their cytotoxic elimination by CD8+ T cells and a delayed onset of neutralising serum antibody titres, as is the case in LCMV infection (PMID: 27812556).

• The RNA sequencing analysis indicates that several genes involved in MHC-II antigen processing and presentation (H2-Aa, CD74, H2-Eb1, H2-Ab1, Ciita) are upregulated in Tfc cells compared to non-Tfc cells. In the methods section the authors note that Tfc cells are sorted by negativity for B220 (to exclude B cells), but this result suggests some B cell contamination may be present in the Tfc population. What are the read counts for these genes in Tfc cells? What was the sort purity for the cell populations in this analysis? The authors should comment on this result.

PLOS authors have the option to publish the peer review history of their article (what does this mean? ). If published, this will include your full peer review and any attached files.

**Do you want your identity to be public for this peer review?** For information about this choice, including consent withdrawal, please see our Privacy Policy .

Reviewer #1: No

Reviewer #2: No

Reviewer #3: No

**Figure resubmission:**

**Reproducibility:**



---

## [Decision Letter · Decision Letter 1]

3 Sep 2025

PPATHOGENS-D-25-00863R1

Follicular CD8+ T cells in Trypanosoma cruzi infection: helpers or killers depending on the target B cell population

PLOS Pathogens

Dear Dr. Gruppi,

Thank you for submitting your manuscript to PLOS Pathogens. After careful consideration, we feel that it has merit but does not fully meet PLOS Pathogens's publication criteria as it currently stands. Therefore, we invite you to submit a revised version of the manuscript that addresses the points raised during the review process.

Please submit your revised manuscript within 30 days Nov 02 2025 11:59PM. If you will need more time than this to complete your revisions, please reply to this message or contact the journal office at plospathogens@plos.org. Please include the following items when submitting your revised manuscript:

We look forward to receiving your revised manuscript.

Kind regards,

Martin Craig Taylor

Guest Editor

PLOS Pathogens

Dominique Soldati-Favre

Section Editor

PLOS Pathogens

Sumita Bhaduri-McIntosh

Editor-in-Chief

PLOS Pathogens

orcid.org/0000-0003-2946-9497

Michael Malim

Editor-in-Chief

PLOS Pathogens

orcid.org/0000-0002-7699-2064

**Additional Editor Comments :**

Please respond to the comments from Reviewer 3 regarding CCR7 and TCF-1 expression and update the plots in Fig 6B.

Note : If the reviewer comments include a recommendation to cite specific previously published works, please review and evaluate these publications to determine whether they are relevant and should be cited. There is no requirement to cite these works unless the editor has indicated otherwise.

**Reviewers' Comments:**

Reviewer's Responses to Questions

**Part I - Summary**

Reviewer #1: The authors have addressed my concerns.

Reviewer #3: See initial review.

**Part II – Major Issues: Key Experiments Required for Acceptance**

Reviewer #1: N/A

Reviewer #3: None.

**Part III – Minor Issues: Editorial and Data Presentation Modifications**

Reviewer #1: N/A

Reviewer #3: -Density plots in figure 6B still require updating to contoured dot plots.

-A few axes for flow plots still require updating with fluorophore labels (for instance: figure 6B, 7D). In figure 3A, for the fluorophore label for CXCR5 on the y-axis, just show which fluorophore is in the representative plot (not both PECy7 and APC).

-Naive CD8 T cells express CCR7 and TCF-1 and since there are naive CD8+ T cells present in the non-Tfc population, the absence of clear CCR7 and TCF-1 positive populations in non-Tfc cells in fig 3C-D is concerning. Have the authors performed CCR7 staining under specific conditions (after resting cells or staining at 37 degrees) which are necessary for CCR7 staining? The authors should compare CCR7 and TCF-1 protein expression in naive CD8+ T cells from non-infected and infected mice to assess whether the apparent absence of CCR7 and TCF-1 protein expression in non-Tfc cells observed here is a result of T cruzi infection or any potential issues with staining these molecules. While I note the response to my previous comment on this issue and data presented in fig S3B, this still needs to be resolved as the absence of CCR7 and TCF-1 from non-Tfc cells (which contain naive CD8+ T cells) is irregular.

PLOS authors have the option to publish the peer review history of their article (what does this mean? ). If published, this will include your full peer review and any attached files.

**Do you want your identity to be public for this peer review?** For information about this choice, including consent withdrawal, please see our Privacy Policy .

Reviewer #1: No

Reviewer #3: No

**Figure resubmission:**

**Reproducibility:**



---

## [Editor Report · Decision Letter 2]

6 Oct 2025

Dear Gruppi,

We are pleased to inform you that your manuscript 'Follicular CD8+ T cells in Trypanosoma cruzi infection: helpers or killers depending on the target B cell population' has been provisionally accepted for publication in PLOS Pathogens.

Best regards,

Martin Craig Taylor

Guest Editor

PLOS Pathogens

Dominique Soldati-Favre

Section Editor

PLOS Pathogens

Sumita Bhaduri-McIntosh

Editor-in-Chief

PLOS Pathogens

orcid.org/0000-0003-2946-9497

Michael Malim

Editor-in-Chief

PLOS Pathogens

orcid.org/0000-0002-7699-2064

The revised manuscript is now acceptable for publication.
---

## [Editor Report · Acceptance letter]

Dear Gruppi,

We are delighted to inform you that your manuscript, "Follicular CD8+ T cells in Trypanosoma cruzi infection: helpers or killers depending on the target B cell population," has been formally accepted for publication in PLOS Pathogens.

Best regards,

Sumita Bhaduri-McIntosh

Editor-in-Chief

PLOS Pathogens

orcid.org/0000-0003-2946-9497

Michael Malim

Editor-in-Chief

PLOS Pathogens

orcid.org/0000-0002-7699-2064